# The N-Myc MB0-MBI region interacts specifically and dynamically with the N-lobe of Aurora kinase A

Johanna Hultman [1], Vivian Morad [1], Eliane Tanner[1], Tristan M. G. Kenney [2,3], Zuzanna Pietras [1], Lalit Pramod Khare[4], Dean Derbyshire[1], Diana Resetca[2,3], Cheryl H. Arrowsmith[2,3,5], Daniel Aili [4], Simon Ekström [6], Linda Z. Penn [2,3], Björn Wallner [7], Alexandra Ahlner [1] ✉ & Maria Sunnerhagen [1] ✉

The intrinsically disordered MYC proteins are master regulators of cellular growth and function, but when deregulated they become cancer drivers. MYC-protein interactions are key to oncogenesis, and while disrupting such interactions would be of significant therapeutic benefit, the intrinsically disordered properties of MYC have dramatically hampered their characterization. Here, we apply an integrated structural biology approach to describe the structure and dynamics of the N-Myc–Aurora A complex, which is critical in neuroendocrine tumor progression. We reveal a functional interaction where multiple binding sites on N-Myc interact with the Aurora A N-lobe. The interaction is governed by aromatic clusters within the conserved MB0 and MBI motifs in N-Myc that interact with Aurora A in a dynamic binding mode that allosterically promotes kinase activation. We show that N-Myc binding to the Aurora A N-lobe can be inhibited by the small-molecule AurkinA, providing opportunity for therapeutical strategies to disrupt this interaction.

The MYC family of proteins (c-Myc, N-Myc, L-Myc) are master transcription factors whose activity is highly regulated as they play key roles in many cellular processes, such as cell differentiation, proliferation, and metabolism. In cancer, multiple mechanisms deregulate MYC expression and activity, leading to uncontrolled cell growth and proliferation. This turns MYC into a potent oncoprotein that significantly contributes to the development of both adult and pediatric human cancers[1–4]. Within the MYC protein family, the N-Myc protein is recognized as a key cancer driver mainly within the central and peripheral nervous system, promoting high-risk cases most often manifested in children. Amplification of the *MYCN* gene has long been used as an indicator of poor prognosis within neuroblastoma[5,6], and characterizes one-in-six cases with dismal prognosis[7].

Directly inhibiting MYC with small molecule inhibitors would mark a key advance in cancer drug development, but this has proven difficult as the MYC proteins are intrinsically disordered in the absence of a protein binding partner[2,8,9]. Intrinsically disordered proteins (IDPs) possess distinct conformational malleability and adaptability, enabling MYC proteins to be highly responsive to regulatory cues and to accommodate versatility in their protein interactions[10,11]. The nature of complexes formed by IDPs spans a continuum, ranging from coupled folding upon binding, over multistate, or "fuzzy", complexes with structured partners, to fully disordered complexes with no persistent structure in either partner[10,12,13]. Within the MYC family, six highly conserved but unstructured regions known as MYC Boxes (MB0-MBIV) have been proposed to serve as focal points for the regulation of

[1]Department of Physics, Chemistry and Biology, Division of Chemistry, Linköping University, Linköping, Sweden. [2]Department of Medical Biophysics, University of Toronto, Toronto, ON, Canada. [3]Princess Margaret Cancer Centre, University Health Network, Toronto, ON, Canada. [4]Department of Physics, Chemistry and Biology, Laboratory of Molecular Materials, Division of Biophysics and Bioengineering, Linköping University, Linköping, Sweden. [5]Structural Genomics Consortium, University of Toronto, Toronto, ON, Canada. [6]BioMS - Swedish National Infrastructure for Biological Mass Spectrometry, Lund University, Lund, Sweden. [7]Department of Physics, Chemistry and Biology, Division of Bioinformatics, Linköping University, Linköping, Sweden. ✉e-mail: alexandra.ahlner@liu.se; maria.sunnerhagen@liu.se

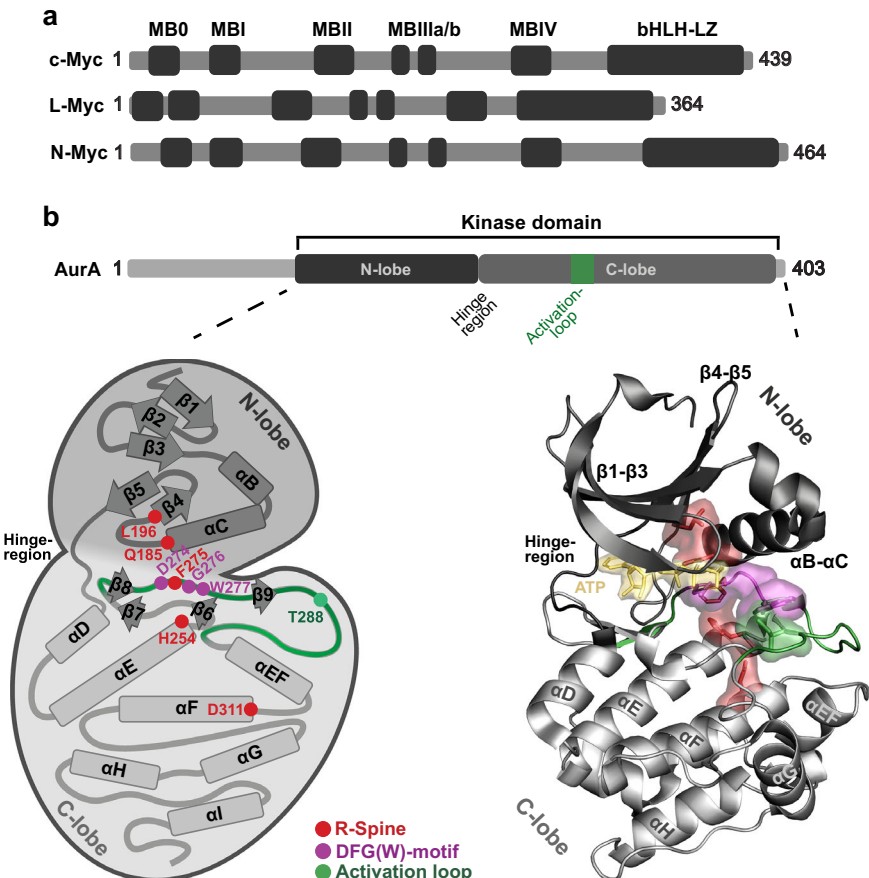

**Fig. 1 | Protein architecture and structure of N-Myc and AurA. a** Protein architecture of MYC family proteins c-Myc, L-Myc and N-Myc, highlighting conserved regions MB0-IV and the bHLH-LZ motif. **b** Protein architecture and crystal structure (PDBID 5DNR) of the Aurora A kinase (AurA) highlighting the kinase domain (residues 122-403) which was used in this study. The kinase domain of AurA adopts a bilobal fold, with the N-lobe (dark gray) and the C-lobe (light gray) connected by a hinge region. Key regulatory features include the activation loop containing the phosphorylation site T288 (green), the DFG(W) motif (purple), and the regulatory spine (red) which together form a regulatory network throughout the protein core. The active site is formed between the N- and C-lobe and contains an ATP-binding site (yellow).

protein interactions[2] (Fig. 1a). Supporting this, an in situ proximity labeling approach revealed 336 high-confidence c-Myc interactors, of which more than half were interrupted by one or more MYC box deletion[14]. Recent advances identifying which of these proteins directly interact with MYC has afforded us and others to reveal structural and dynamic features of protein interactions to MYC boxes and adjacent residues[15–22], several of which show characteristic IDP recognition modes[10,12,23]. With these insights, alternative opportunities to inhibit MYC oncogenic activity could be made possible by disrupting MYC interactions with key partner proteins.

Within N-Myc driven neuroblastoma, the Aurora A (AurA) protein kinase is upregulated and stabilizes cellular N-Myc levels in a post-transcriptional manner, presumably as part of an oncogenic mechanism[24]. Aurora kinases are widely overexpressed in tumors, with specific biochemical roles identified for AurA in melanoma, neuroblastoma and prostate cancer[25,26]. AurA shares a common regulatory architecture with other eukaryotic protein kinases, comprising a bilobal kinase domain with the active site positioned between its N- and C-lobes[25,27] (Fig. 1b). A regulatory spine spans four layers of interacting side chains, including the catalytically critical DFG(W) motif at the active site. This network enables allosteric kinase activation by connecting the active site with the N-lobe αC-β4 loop and the activation loop, which contains an activating phosphorylation site (Fig. 1b)[27]. By crystallographic studies, the spindle assembly factor TPX2 was shown to activate AurA by binding to hydrophobic pockets in the N-lobe, thereby affecting the adjacent αC

helix[28]. However, later studies suggest that Aurora kinases in solution comprise an ensemble of states, and that their activation is conveyed through the regulatory spine by induced population shifts[29]. Furthermore, an expanding body of research highlights the role of correlated motions and dynamics-based allostery in regulating catalytic turnover in kinases[30,31].

Inhibiting the N-Myc–AurA interaction has been suggested as a therapeutic strategy for childhood cancers[9]. Several AurA inhibitors promote proteasomal breakdown of N-Myc but were cytotoxic in subsequent clinical trials[32–34], prompting further investigation of the N-Myc–AurA complex. While it has been suggested that larger regions of the N-Myc transactivation domain (TAD) bind AurA[35], the crystal structure of AurA together with N-Myc$_{28-89}$ only resolved N-Myc$_{61-89}$, in a position supporting the AurA activation loop in its open state[18]. This prompted the working model that the N-Myc$_{61-89}$–AurA interaction protects N-Myc from degradation via the E3 ligase SCF$^{FbxW7}$ pathway[18] and subsequent work has thus been focused on this N-Myc region[35–37]. However, while N-Myc$_{28-89}$ was able to activate AurA, N-Myc$_{61-89}$ alone had no effect on AurA function[18] and constraining the conformational space of N-Myc$_{61-89}$ by shorter cyclic peptides did not significantly enhance inhibitory potency of the N-Myc–AurA interaction[36]. Taken together, this suggests that the full molecular mechanism describing the N-Myc–AurA functional interplay has not yet been resolved.

In this work, we show that N-Myc binds AurA in a 1:1 complex via several contact regions on N-Myc. Among these, the N-terminal MB0

and MBI of N-Myc jointly comprise primary binding regions, with distinct hydrophobic patches that bind and stabilize the AurA N-lobe, thereby enhancing AurA activity. Using an experimentally integrated AlphaFold-based modeling approach, we derived a conformational ensemble model for the N-Myc-AurA interaction, consistent with experimental data from multiple structural and biophysical techniques in solution, including nuclear magnetic resonance spectroscopy (NMR), small angle X-ray scattering (SAXS), hydrogen deuterium exchange mass spectrometry (HDX-MS), nano differential scanning fluorimetry (nano-DSF), isothermal titration calorimetry (ITC), biolayer interferometry (BLI), dynamic light scattering (DLS) and size-exclusion chromatography with multi-angle light scattering (SEC-MALS). In the context of an intact N-Myc segment encompassing both MB0 and MBI (N-Myc$_{1-100}$), the previously identified binding region N-Myc$_{61-89}$[18] is a minor contributor to AurA binding and activation and does not alter the conformational ensemble. We propose that N-Myc binding via MB0 and MBI to the AurA N-lobe can be targeted as an alternative therapeutic route.

## Results

### AurA forms 1:1 complexes with both N-Myc$_{1-69}$ and N-Myc$_{1-100}$

To understand how the N-terminal region of N-Myc binds AurA in solution we produced N-Myc$_{1-69}$ and N-Myc$_{1-100}$ from TEV-cleavable His-TRX fusion proteins, adding only one additional Ser at their N-termini. Both constructs contain the entire MB0 and MBI regions, which are well conserved in the MYC family, known hubs for protein interactions, and functionally important for MYC oncogenesis[14] (Fig. 2a, Supplementary Fig. 1a). The longer N-Myc$_{1-100}$ construct also

includes the non-conserved region C-terminal to MBI (N-Myc$_{70-89}$) which co-crystallized with AurA$_{122-403}$[18]. We produced AurA$_{122-403}$ including previously described mutations C290A and C393A[38] but with a cleavable SUMO tag, resulting in protein with no non-native N-terminal residues and in sufficient yields for us to probe the N-Myc interaction with the kinase domain of AurA.

All constructs were monomeric as judged by SEC-MALS, each eluting as a single peak with respective average molecular masses in agreement with a monomeric state (Fig. 2b, Supplementary Fig. 1b). By DLS, the calculated hydrodynamic radii for N-Myc$_{1-69}$ and N-Myc$_{1-100}$ were significantly larger and had more variability than expected for a globular protein[39] (Supplementary Fig. 1c). This agrees with the longer-than-expected retention times observed in the SEC elution profiles and suggests a high degree of intrinsic disorder. This was further supported by SAXS, where smooth I(q) scattering profiles suggest an averaging of multiple conformational states and the plateau at large angles in the dimensionless Kratky plot indicates a natively unfolded state (Supplementary Fig. 2). Furthermore, the maximum dimensions (D$_{max}$) of the two N-Myc constructs as derived from the pair-distance distribution function analysis (P(r)) are large, in agreement with the DLS results (Supplementary Table 1)[40].

To characterize stoichiometry and affinities of the N-Myc–AurA interaction, we used SEC-MALS and SAXS together with ITC. Judging by the clear shift in elution volume and SDS-PAGE analysis, both N-Myc$_{1-69}$ and N-Myc$_{1-100}$ eluted as complexes with AurA on a MALS coupled SEC-column (Fig. 2c, Supplementary Fig. 3a). SEC-SAXS independently confirmed 1:1 stoichiometry of N-Myc–AurA complexes by Guinier analysis and molecular weight estimation. In addition, P(r) analysis

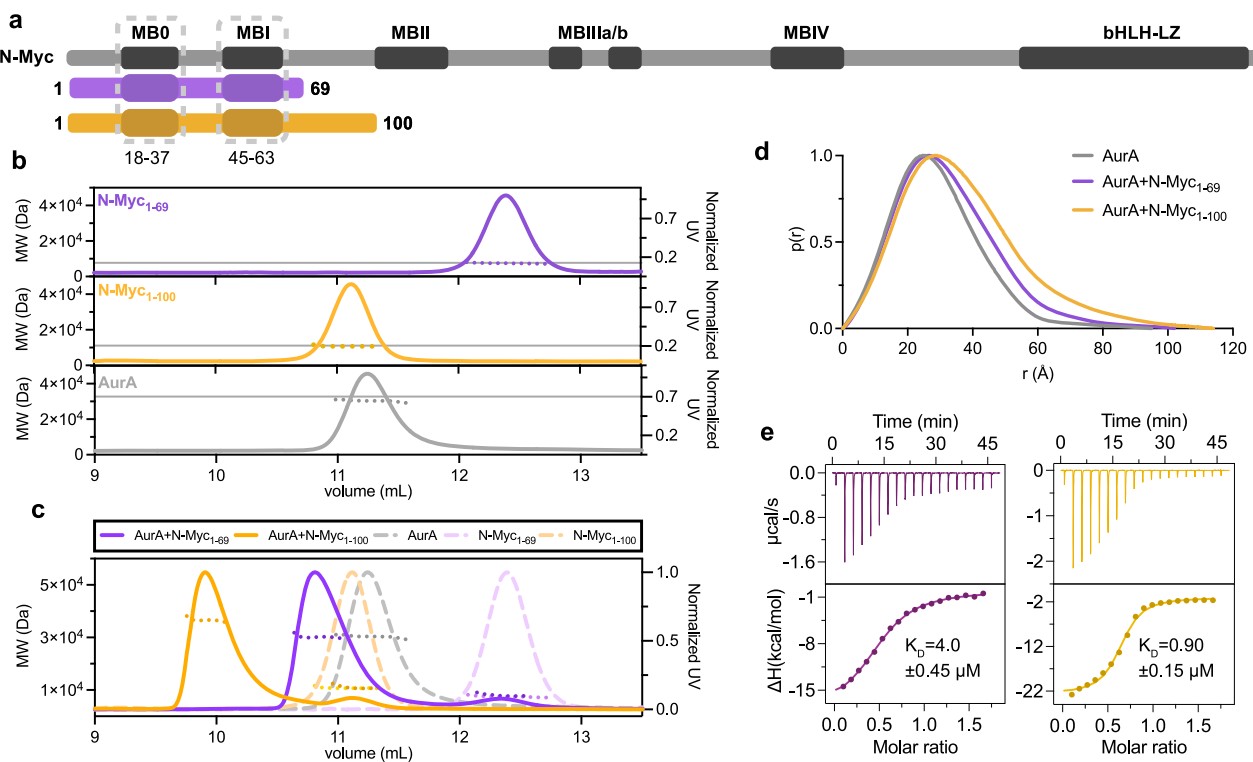

**Fig. 2 | Biophysical characterization of the N-Myc–AurA complex. a** Schematic representation of the two N-Myc constructs used in this study (purple and gold) shown in the context of the entirety of the N-Myc protein architecture (gray). Residue boundaries are given for MB0 and MBI. **b** SEC-MALS elution profiles for N-Myc$_{1-69}$ (purple), N-Myc$_{1-100}$ (gold) and AurA (gray), with the respective calculated (dotted lines) and theoretical (solid lines) molecular weights (7.47 kDa (± 6%), 11.2 kDa (± 4%) and 30.4 kDa (± 0.4%), compared to the respective theoretical molecular weights of 7.68, 11.1 and 32.7 kDa for N-Myc$_{1-69}$, N-Myc$_{1-100}$ and AurA). **c** Overlay of SEC-MALS elution profiles for individual proteins (dashed lines) and their complexes (solid lines), illustrating complex formation. **d** Paired distance distribution function P(r) derived from SEC-SAXS measurements for AurA alone and in complex with N-Myc$_{1-69}$ or N-Myc$_{1-100}$, with estimated D$_{max}$ at 95, 102 and 114 Å respectively. **e** ITC curves for AurA binding to N-Myc$_{1-69}$ and N-Myc$_{1-100}$, and the calculated K$_D$ values of 4.0 ± 0.45 μM and 0.90 ± 0.15 μM, respectively (see also Supplementary Table 2).

indicated an increase in overall dimension upon complex formation compared to AurA alone, while Kratky analysis revealed predominantly globular conformation with some retained flexibility, as evidenced by the bell-shaped curve and upturn at higher q-values (Fig. 2d, Supplementary Fig. 3b, c, Supplementary Table 1). By ITC, the dissociation constants ($K_D$) were $4.0 \pm 0.45$ µM and $0.90 \pm 0.15$ µM for N-Myc$_{1-69}$ and N-Myc$_{1-100}$ respectively (Fig. 2e, Supplementary Table 2), which is consistent with $K_D$s obtained for N-Myc$_{28-89}$ by other methods[18], and for c-Myc TAD binding to regulatory partners[16,17,21]. While ITC revealed a slightly higher affinity to AurA for the longer N-Myc construct, the similar $K_D$s indicate that the main binding determinants are contained within N-Myc$_{1-69}$, whereas additional residues in N-Myc$_{1-100}$ contribute to binding, but are not solely responsible. All ITC data were evaluated using a 1:1 binding model, in agreement with the stoichiometry identified by SAXS and SEC-MALS. With this model the N-value was consistently lower than 1, suggesting additional binding sites on the titrand that contributes to early saturation[41]. This would be consistent with co-elution on a SEC column, as multiple interaction sites could maintain complex formation despite relatively low affinities. The fitted ITC data further revealed similar thermodynamic profiles, suggesting similar binding modes to AurA for the two N-Myc constructs (Supplementary Table 2).

## Three distinct N-Myc regions interact with AurA

To learn more about the N-Myc–AurA interaction, we proceeded to NMR. [1]H[15]N-HSQC of N-Myc$_{1-69}$ and N-Myc$_{1-100}$ confirmed their intrinsically disordered nature, as evidenced by narrow linewidths and backbone amide proton chemical shifts confined to the 7.4–8.6 ppm range, consistent with corresponding NMR analysis of similar c-Myc and N-Myc fragments[16,20,21,35]. The superposition of the N-Myc$_{1-69}$ and N-Myc$_{1-100}$ spectra is nearly perfect, suggesting only minor, if any, internal interactions between N-Myc residues 70-100 and regions within N-Myc$_{1-69}$ (Supplementary Fig. 4a, b). Chemical shift secondary structure population inference (CheSPI) analysis[42] further confirmed that the two N-Myc constructs are largely disordered, with only slight, possibly transient, helicity proposed within MBI and in residues 75-86 in N-Myc$_{1-100}$ (Supplementary Fig. 4c).

To identify specific residue motifs within N-Myc that interact with AurA in solution, we titrated unlabeled AurA into [13]C[15]N-labeled N-Myc$_{1-69}$ and N-Myc$_{1-100}$. The resulting NMR spectra revealed three distinct amino acid regions exhibiting pronounced line broadening (reduced I/I$_0$) even at substoichiometric concentrations of AurA, accompanied by minor chemical shift perturbations (CSPs) (Fig. 3). This pattern is characteristic of interactions between IDPs and globular proteins[43–47] and has also been reported for several c-Myc

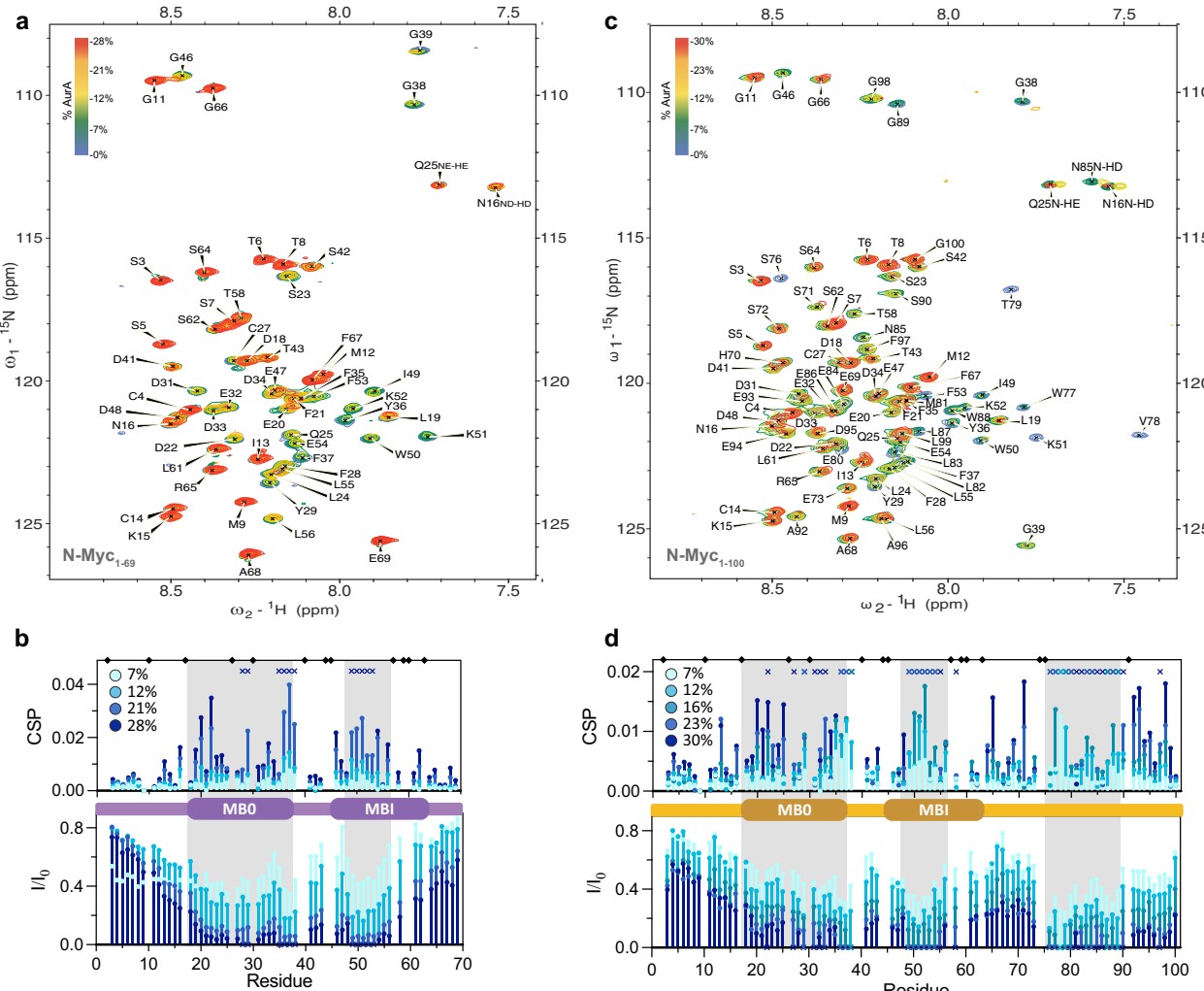

**Fig. 3 | AurA imprint on N-Myc by NMR.** Titration of unlabeled AurA to isotopically labeled **a,b** N-Myc$_{1-69}$ and **c,d** N-Myc$_{1-100}$, respectively. For each construct, [1]H[15]N HSQC spectra are shown in overlay across titration points with 0-30% molar equivalents of AurA, relative to N-Myc. Bar graphs display chemical shift perturbations (CSP) and relative signal intensity changes (I/I$_0$) at each titration point.

Regions exhibiting significant signs of binding (residues 19–38 (MB0), 48–55 (in MBI) and 76–89) are highlighted in gray. Prolines, which are undetectable in the HSQC spectra, are marked with black diamonds. Signals that disappear below the noise threshold due to line broadening are indicated with an "x" in the corresponding color.

interactions[16,17,21]. The affected regions in N-Myc include the entire MB0 (residues 18-38), residues 48-55 within MBI (both N-Myc$_{1-69}$ and N-Myc$_{1-100}$), and residues 76-89 C-terminal to MBI (in N-Myc$_{1-100}$). The most prominent, but still small, CSPs were centered around aromatics in MB0 (F21, F28, Y29, F35, Y36, F37) and in MBI (W50, F53), including charged residues D20 and D22 which flank F21, and K51 and K52 located between W50 and F53. In N-Myc$_{1-100}$, the third region affected by AurA includes aromatic residues W77 and W88, as well as adjacent residues, and encompass the structured N-Myc binding site previously identified by crystallography[18]. In solution, the CSPs and I/I$_0$s in this region closely resemble those observed in MB0 and MBI. Importantly, the similar NMR binding pattern for MB0 and MBI residues in both N-Myc$_{1-69}$ and N-Myc$_{1-100}$ suggests that AurA binds MB0 and MBI independent of the N-Myc C-terminal segment beyond MBI. This agrees well with our ITC, SEC-MALS and SAXS results suggesting that both N-Myc$_{1-69}$ and N-Myc$_{1-100}$ form 1:1 AurA complexes and bind AurA with comparable affinities.

The detected line broadening in N-Myc upon AurA addition may arise from an enhanced observed transverse relaxation ($R_2$) due to one or more of the following factors: i) the increased molecular size of the complex resulting in a longer correlation time, ii) conformational exchange on the μs-ms time scale between free and bound states, and/or iii) exchange processes in the bound state due to transient intra- and intermolecular interactions[47]. As peak intensities for N-Myc could not be recovered at >90% saturation (Supplementary Fig. 5), we speculate that transverse relaxation in the bound state is much larger than that of free N-Myc, indicating partial restriction of its intrinsically disordered nature. Furthermore, considering the expected high relaxation in the bound state and the extent of signal loss due to line broadening exceeding the proportion of added AurA in our substoichiometric titrations, we conclude that the system undergoes conformational exchange at a rate that is faster than the transverse relaxation of free N-Myc[48].

To further explore the interaction profile and dynamics of the N-Myc$_{1-69}$–AurA complex, we determined fast relaxation parameters; longitudinal relaxation rates ($R_1$), transverse relaxation rates ($R_2$) derived from rotating frame relaxation rates ($R_{1\rho}$) and $R_1$s, as well as heteronuclear NOEs (hetNOEs) of N-Myc$_{1-69}$, both in isolation and in the presence of increasing substoichiometric concentrations of AurA (Fig. 4). Consistent with the observed intensity changes, the $^{15}$N $R_2$ rates increased significantly across the MB0 region and within a narrower segment within MBI, while remaining low for terminal residues and within the linker between MB0 and MBI, indicating that the MB0 and MBI regions are specifically involved in the interaction with distinct dynamic profiles. In agreement with the CSPs, the largest increases in $^{15}$N $R_2$ rates were observed for F35, Y36 and F37 in MB0, and I49, W50 and K51 in MBI, where these resonances were broadened beyond detection at addition of only 10% AurA. In contrast to the observed changes in $R_2$ rates, $R_1$ remained largely unaffected by the addition of AurA. This observation is reasonable even if the free and bound states differ in correlation time or intrinsic dynamics, since $R_1$ rates are in general small for $^{15}$N in biomolecular systems[48]. Furthermore, hetNOE values remained largely unchanged upon addition of AurA, suggesting no significant increase in backbone rigidity upon complex formation. However, it is possible that the low concentration of AurA was insufficient to reveal subtle dynamic changes. Concluding our NMR analysis, three distinct regions in N-Myc are similarly perturbed by AurA binding, as shown by CSPs, line broadening, and increased $R_2$ values. The loss of signal at substoichiometric AurA concentrations likely reflects conformational exchange on the μs-ms time scale and a system with substantial differences in transverse relaxation rates.

## Hydrogen-deuterium exchange shows that N-Myc$_{1-69}$ and N-Myc$_{1-100}$ both bind to the AurA N-lobe

As poor NMR resolution of AurA has hitherto precluded direct NMR analysis[29], we turned to HDX-MS to investigate the effects in AurA upon

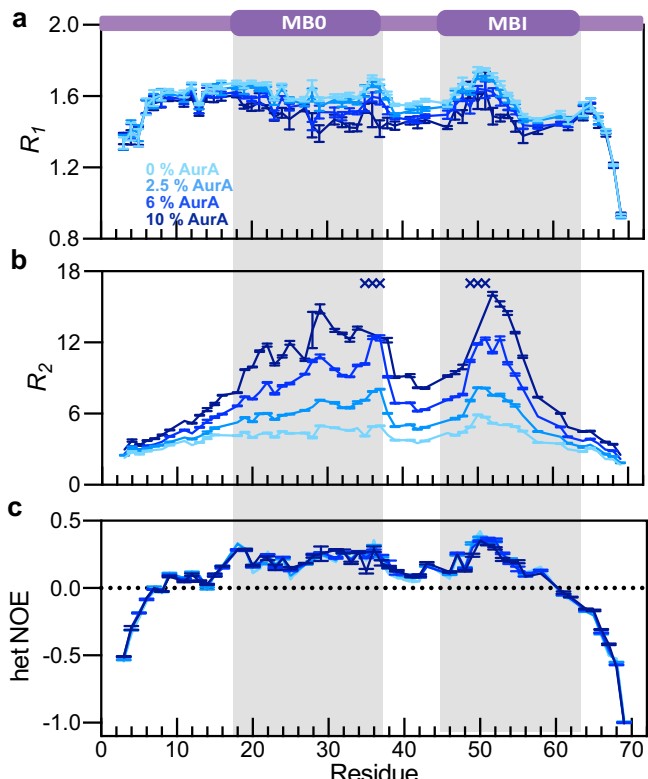

**Fig. 4 | Fast relaxation parameters of N-Myc$_{1-69}$ upon titration with Aurora A.**
**a** Longitudinal relaxations rates ($R_1$) **b** Transverse relaxations rates ($R_2$) and **c** heteronuclear NOE (hetNOE) values for $^{15}$N labeled of N-Myc$_{1-69}$ measured by NMR spectroscopy in the presence of increasing substoichiometric concentrations of AurA (0%, 2.5%, 6% and 10% molar equivalents). Disappearance of signal below the noise level due to line broadening at specific titration points is indicated by an "x" in the corresponding color. Error bars were calculated based on three sets of replicate measurements using the jackknife method embedded within PINT.

N-Myc binding in solution. Amides in well-ordered and/or buried regions exchange more slowly compared to those in flexible and/or solvent-exposed regions, rendering this technique suitable both to explore changes in internal dynamics and monitor protection of exposed surface areas by an interacting protein or ligand[49]. To probe the effects of N-Myc binding on AurA, we compared the HDX patterns of AurA alone to that of AurA in complex with N-Myc$_{1-69}$ and N-Myc$_{1-100}$ at four time points: 30, 300, 3000 and 9000 s (Supplementary Data 1-2). To enable comparative analysis of the N-Myc$_{1-69}$ and N-Myc$_{1-100}$ binding across the two distinct HDX-MS runs, we selected 96 AurA peptides with high significance and reproducibility in both experiments, resulting in a common peptide pool representative for all regions in AurA (Fig. 5a, b, Supplementary Data 3; see Methods).

The AurA HDX pattern revealed significantly reduced H/D exchange upon N-Myc binding. The most pronounced effects were localized in the AurA N-lobe, while the C-lobe remained comparatively unaffected (Fig. 5a). The AurA HDX patterns obtained from N-Myc$_{1-69}$ and N-Myc$_{1-100}$ binding were surprisingly similar and no additionally protected regions in AurA were observed for N-Myc$_{1-100}$ (Fig. 5b), even though our NMR and ITC data suggest that this longer construct holds a third AurA binding site. Rather, minor differences were observed only in the magnitude of the protection (Supplementary Fig. 6a), where the higher affinity of N-Myc$_{1-100}$ compared to N-Myc$_{1-69}$ enable a greater degree of saturation under the experimental conditions (87% compared to 69%, see Methods for details), potentially leading to an underestimation of the protection conferred by N-Myc$_{1-69}$ binding. Within the AurA C-lobe, residues 334-351, encompassing αG which is contacted by N-Myc-W77 and -W88 in the N-Myc–AurA crystal

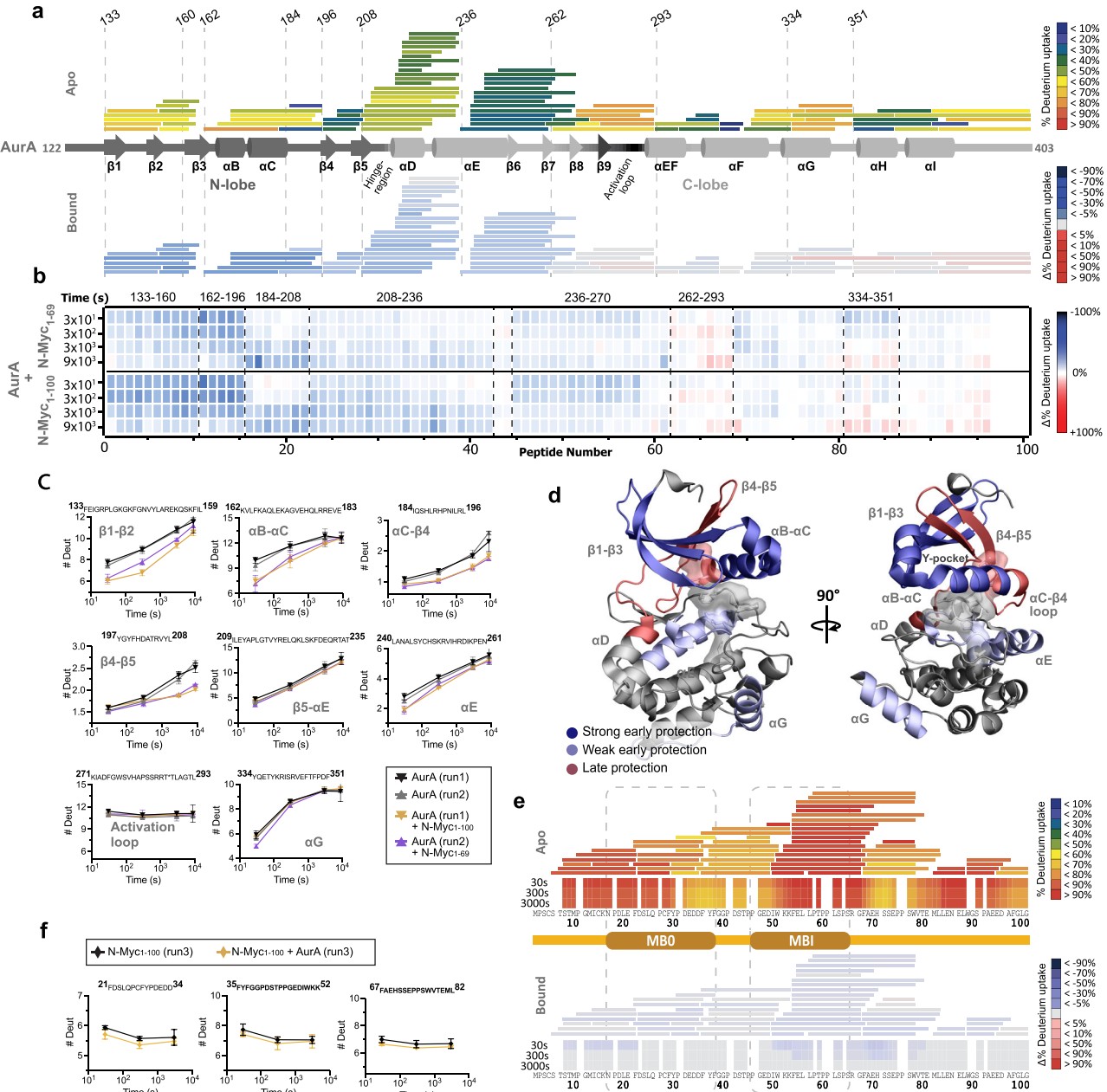

**Fig. 5 | N-Myc imprint on AurA by HDX-MS. a** AurA peptides (bars) from HDX-MS, aligned with the structural architecture of AurA to show the sequence coverage. Peptides were selected by filtering based on high significance and reproducibility between HDX-MS run1 and run2 to generate this common peptide pool, allowing for direct comparison between runs. Bars are colored after the average deuterium uptake over all investigated timepoints (30 s, 300 s, 3000 s and 9000 s) for unbound AurA (top), and the average differential deuterium uptake for each AurA peptide in the presence of N-Myc$_{1-100}$ (bottom) over all measured timepoints. **b** Chiclet plot showing the differential protection for all AurA peptides in the common pool upon addition of N-Myc$_{1-69}$ (top) and N-Myc$_{1-100}$ (bottom) across all measured time points. **c** Uptake plots for a representative set of peptides, showing the mean number of exchanged deuterons and standard deviations from triplicate measurements at the four measured timepoints for N-Myc$_{1-100}$-bound AurA (gold), N-Myc$_{1-69}$-bound AurA (purple), and AurA alone (black and gray, respectively referencing the N-Myc$_{1-100}$ (run1)/ N-Myc$_{1-69}$ (run2) experiments). The sequence for

each peptide fragment is shown above the plot, and AurA structural elements encompassed by each peptide in the folded structure are annotated in gray. **d** AurA structure (PDBID: 5DNR) with representative coloring for the observed HDX-pattern: direct binding as seen by early protection at one (weak; light blue) or several (strong; dark blue) timepoints, and decreased dynamics or allosteric effects as seen by late protection (pink). Only HDX effects persistent in several peptides in the same region were mapped onto the AurA structure. Regulatory residues (R-spine and DFG(W)-motif) are highlighted in the structure, with coloring representative for the HDX effects for that region. **e** Peptides (bars) for apo and AurA bound N-Myc$_{1-100}$ during HDX-MS, aligned with the schematic sequence of N-Myc, including MB0 and MB1, to show the HDX sequence coverage. Bars are colored as described in (**a**). **f** Uptake plots for a representative set of N-Myc peptides, showing the mean number of exchanged deuterons and standard deviations from triplicate measurements at three measured timepoints (30 s, 300 s, 3000 s) for AurA bound N-Myc$_{1-100}$ (gold) and N-Myc$_{1-100}$ alone (black).

structure[18], showed only a slight reduction in H/D exchange at the earliest time point for N-Myc$_{1-69}$ and N-Myc$_{1-100}$ (Fig. 5a–c, Supplementary Fig. 6a). While a highly transient interaction comprising residues within N-Myc$_{70-100}$ and αG in AurA might not have been resolved

in the HDX-MS experiment, a direct comparison of the magnitude of the HDX effects across the AurA sequence supports that the AurA N-lobe is indeed the main N-Myc binding region (Supplementary Fig. 6a). Furthermore, the similar HDX protection pattern on AurA

observed for both N-Myc$_{1-69}$ and N-Myc$_{1-100}$ suggests that N-Myc$_{1-69}$ houses the main AurA interaction site.

We further analyzed the time-dependence of the AurA HDX pattern and compared this with known dynamic features in kinases. The largest cumulative difference in H/D exchange over all time points was observed within residues 133-196 in the AurA N-lobe (Fig. 5b, Supplementary Fig. 6a). Within this region, residues 162-184 encompass helices αB and αC, which together with the tip of β3 line the so-called "Y-pocket", a well-known regulatory binding groove in kinases[50]. Here, the AurA HDX pattern reveals significantly reduced H/D exchange upon N-Myc binding at short D$_2$O exposure times, but at longer time points the H/D exchange becomes similar for the bound and free state (Fig. 5b, c). Such early protection features are commonly observed at direct interaction sites by transient, dynamic, or fuzzy binders, where rapid on- and off-rates allow for continued H/D exchange in the bound state[51]. To visualize the protected areas on the AurA structure (Fig. 5d), we classified significant protection at early time points as 'strong' if it persisted over several of the time points measured, and 'weak' if it persisted over only one time point. Residues 184-208 include the AurA N-lobe αC-β4 loop, a functional and well-recognized element within protein kinases that structurally connects the N- and C-lobe[25,27,52], and the β4-β5 region. In contrast to the early protection observed around the Y-pocket, N-Myc binding conveys significant late protection within residues 184-208 at longer time points (Fig. 5b, c), as visualized in Fig. 5d. This suggests a reduction of conformational dynamics in this already well-protected region, which includes highly conserved residues such as Q185 and L196 of the regulatory spine, thereby connecting to the AurA active site. A weak initial HDX protection pattern was also observed at residues 236-262, comprising part of the C-lobe αE (Fig. 5b, c), which connects to the AurA active site via R-spine residue H254 and packs against the AurA αC-β4 loop via Y246-H190 stacking interactions[52].

To investigate the H/D exchange on N-Myc in the AurA-bound state, we performed an N-Myc optimized experiment at 1:1 molar ratio (77% saturation) using enhanced proteolytic resolution (Fig. 5e, Supplementary Data 4-5). This setup contrasts the AurA-optimized HDX-MS experiments, which were performed at 3-fold N-Myc excess to maximize AurA saturation and could therefore not resolve N-Myc protection in the bound state. While the same pattern of large HDX perturbations were observed for AurA (ΔHDX 1-3 Da), indicating high occupancy of N-Myc at the AurA binding sites, only very weak perturbations were detected for N-Myc at the shortest time points (ΔHDX 0.2-0.6 Da), where the deuterium uptake was already high across the N-Myc sequence (Fig. 5f, Supplementary Fig. 6b). This suggests that N-Myc remains largely disordered upon binding and does not adopt any stable secondary structure. These results distinguish the N-Myc–AurA interaction from the INCENP-Aurora B interaction, where HDX-MS studies revealed folding-on-binding of a highly protected INCENP peptide[53]. Furthermore, N-Myc binding at the Y-pocket reduces the conformational dynamics in the αC-β4 loop, a change which may propagate into the active site via the regulatory spine and thus affect AurA activity.

## N-Myc binding is independent of AurA activation loop phosphorylation

In the HDX-MS experiment, AurA peptides encompassing the AurA activation loop were equally poorly protected in the absence and presence of N-Myc (Fig. 5a–d), suggesting maintained activation loop dynamics of AurA in complex with the N-Myc constructs investigated here. The MS/MS evaluation of our *E. coli* expressed AurA construct performed within the HDX-MS-setup showed phosphorylation in at least one site in the AurA activation loop (Supplementary Fig. 7a), in agreement with previous observations[28], and to increase sequence coverage these modified peptides were therefore included in the HDX-MS analysis. Comparing the H/D exchange for phosphorylated and

unphosphorylated peptides within the activation loop revealed identical patterns for both groups (see deposited HDX-MS data to the PRIDE repository). However, to further evaluate whether N-Myc binding to AurA is dependent on the phosphorylation status of the activation loop, we expressed AurA with mutations T287A:T288E to mimic the activating phosphorylation at T288 in the activation loop while prohibiting phosphorylation at the adjacent T287. We also co-expressed AurA with λ-phosphatase[54], resulting in AurA with no phosphorylations. Binding analysis monitored by BLI using these AurA variants revealed no significant changes in affinities for N-Myc binding based on the AurA phosphorylation status (Supplementary Fig. 7b). This suggests that N-Myc binding is not affected by phosphorylation in the AurA activation loop, once again in contrast to the INCENP interaction with Aurora B, which is highly stabilized by Aurora B phosphorylation[53].

## N-Myc MB0 stabilizes and activates AurA

To investigate whether N-Myc binding modulates AurA thermal stability, we monitored the thermal melting point ($T_m$) by nanoDSF[55]. The AurA tryptophan (Trp) residues in the N-lobe (W128), in the C-lobe (W313, W382), and within the DFGW motif at the active site (W277) are well positioned to monitor global unfolding. N-Myc$_{1-69}$ has one Trp (W50) and N-Myc$_{1-100}$ another two (W77, W88). Neither N-Myc$_{1-69}$ nor N-Myc$_{1-100}$ showed any sign of cooperative melting (Fig. 6a), as expected for an IDP. Still, both N-Myc constructs significantly stabilized AurA to similar extents, and in a concentration dependent manner in line with their respective $K_D$s as observed by ITC. In contrast, addition of peptides N-Myc$_{61-90}$ or N-Myc$_{69-90}$, corresponding to the third AurA interaction site in N-Myc$_{1-100}$, resulted in a slightly decreased $T_m$ compared to free AurA (Fig. 6a,b). This effect could be due to an excess of unbound N-Myc, yielding an increase in the 350 nm signal and a masking of the unfolding effect (Supplementary Fig. 8). Indeed, a slight increase was observed in the onset ($T_{ON}$) of the AurA melting curve, but this increase was still much lower than that observed for N-Myc$_{1-69}$ and N-Myc$_{1-100}$ (Fig. 6b). Taken together, these results support that the main N-Myc determinants for AurA interaction reside within N-Myc$_{1-69}$.

To assess possible N-Myc anchor points for binding AurA, we designed alanine mutations in N-Myc$_{1-69}$ guided by our NMR analysis. We focused on the aromatic residues and clusters affected by AurA binding and analyzed the effects on $T_m$ in the respective N-Myc–AurA complexes. Mutation of either of the two aromatic clusters in MB0 (F28A:Y29A or F35A:Y36A:F37A) significantly reduced thermal stabilization of AurA, as judged from the onset and inflection points of the complex melting curve (Fig. 6b, Supplementary Fig. 8). Substitution of W50A within MBI had similar effects, however not to the same extent. The smallest effects were seen for mutants F53A and F21A, which only marginally affected the stability compared to wild type N-Myc.

To assess whether reduced stability of the N-Myc–AurA complex correlates with binding affinity, we performed ITC measurements using N-Myc anchor point mutants. While all mutants exhibited reduced affinity for AurA, the aromatic cluster mutations within MB0 showed the most pronounced effects (Fig. 6c, d, Supplementary Fig. 9, Supplementary Table 2). ITC measurements further indicated $K_D$s exceeding 100 μM for the N-Myc$_{69-90}$ peptide, suggesting that this region alone cannot bind AurA efficiently. The magnitude of the affinity reduction seen by ITC correlates well with the diminished thermal stabilization observed by nanoDSF (Fig. 6e) supporting a correlation between binding strength and complex stability.

We continued to deconstruct the functional components of N-Myc binding by performing AurA activation assays in the presence and absence of the designed N-Myc variants using the ADP-Glo Kinase Assay (Promega). The results showed that binding of N-Myc$_{1-69}$ or N-Myc$_{1-100}$ increases the kinase activity of AurA (Fig. 6f). In contrast, N-Myc peptides 61–90 and 69–90 did not have any significant effect on

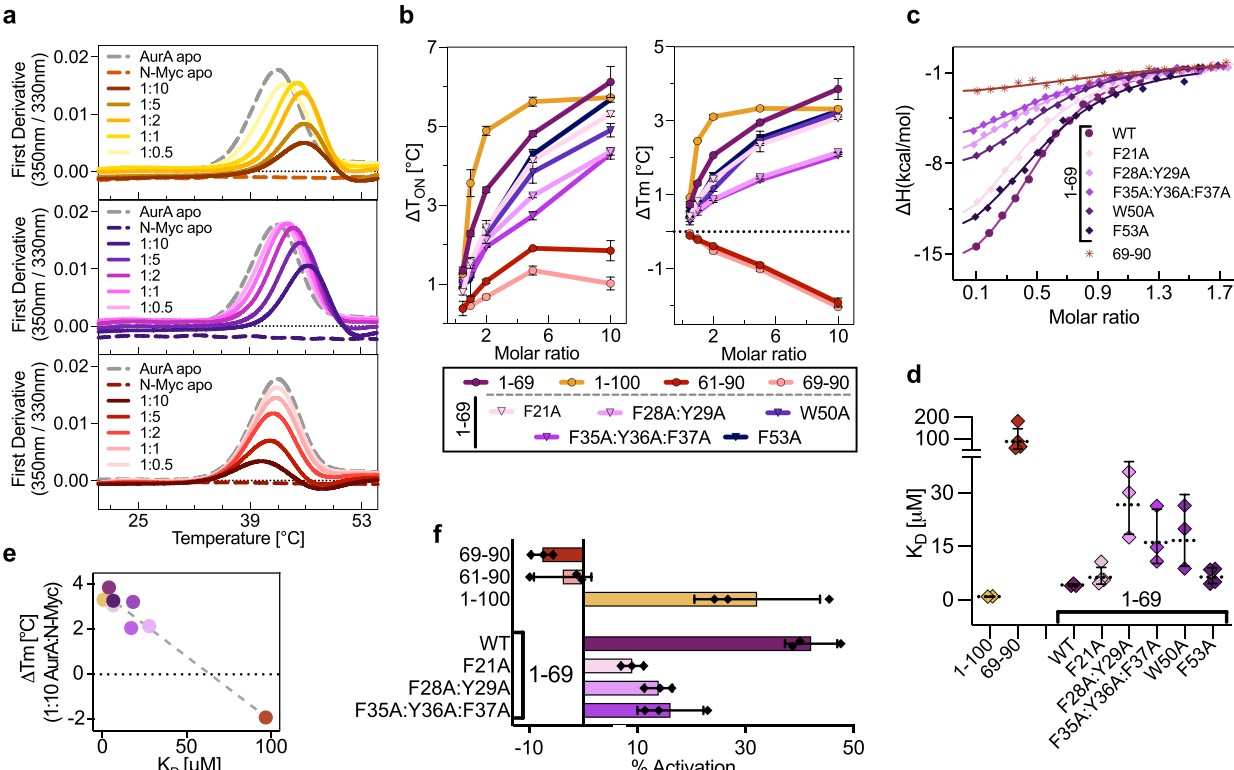

**Fig. 6 | Mutations of aromatic residues effect N-Myc–AurA binding. a** AurA thermal stability as a function of N-Myc concentration (0, 10, 20, 100 and 200 µM) for N-Myc$_{1-69}$, N-Myc$_{1-100}$ and N-Myc$_{69-90}$ monitored by nanoDSF via the first derivative of the 350 nm/330 nm absorbance signal. **b** Differences in melting onset ($\Delta T_{ON}$) and thermal melting point ($\Delta T_m$) of the AurA melting curve in the presence of N-Myc variants of increasing concentrations (0, 10, 20, 100 and 200 µM) measured by nanoDSF. Measurements were done in duplicates and the mean and standard deviations are shown. **c** ITC data for N-Myc variants binding to AurA and

**d** binding affinities ($K_D$) derived from triplicate measurements for each interaction (see also Table S2), showing mean (horizontal line) and standard deviation. **e** $\Delta T_m$ for the AurA–N-Myc complexes at 10-fold excess N-Myc (200 µM) derived from nanoDSF, shown as a function of $K_D$ derived from ITC for the N-Myc construct and mutants ($R^2 = 0.95$) (**f**) AurA kinase activity, shown as the percent increase observed in the presence of various N-Myc constructs and variants compared to apo AurA. Measurements were done in triplicates and mean, indicated by the respective bars, and standard deviation from mean (error bars) are shown.

AurA activity. Alanine mutations of aromatic clusters F28:Y29 or F35:Y36:F37 within MB0 significantly decreased the activating effect by N-Myc$_{1-69}$, in line with the importance of these clusters for binding and stability. Additionally, the MB0 F21A mutant resulted in similar reduction in activation, despite its limited effects on binding affinity and stabilization of AurA. These results indicate that aromatic residues in the N-Myc N-terminal region hold a critical functional role in the interaction with AurA.

## A multistate complex model for N-Myc–AurA interactions in solution

To put our experimental data in a structural context, we turned to computational methods. Initial attempts to use contact information from NMR and/or HDX-MS to derive models of the complex had limited success, as such models were generally too compact to fit our SAXS data. To allow for more extensive sampling of the conformational space, we instead generated 10,000 models using AFsample[56], an improved AlphaFold sampling algorithm. Two versions of the AlphaFold-multimer neural networks (multimer_v1, and multimer_v2) were used without incorporating any templates to generate models of AurA$_{122-403}$ in complex with N-Myc$_{1-69}$ and N-Myc$_{1-100}$, respectively. These models were then filtered based on their quality of fit to the experimental SAXS scattering profiles for the respective complexes by fitting each model using the Pepsi-SAXS software[57]. Structures generated using multimer_v1 yielded relatively poor $\chi^2$ when fitted to the experimental SAXS data compared to those generated by multimer_v2, despite the average ranking confidence being better (Supplementary

Fig. 10a). When comparing the generated models, we noted a tendency of multimer_v1 to preferentially generate tightly packed complexes with smaller radius of gyration, thus disfavoring variability. Indeed, when selecting models that fit the experimental SAXS data ($\chi^2 < 1.1$) to structurally describe the conformational ensembles of N-Myc$_{1-69}$ and N-Myc$_{1-100}$ when bound to AurA, only models generated by multimer_v2 were obtained (Supplementary Fig. 10b). Further refinement by including additional structures with higher $\chi^2$ did not result in any lowering of the $\chi^2$ of the ensemble.

Overall, the N-Myc$_{1-69}$–AurA and N-Myc$_{1-100}$–AurA conformational ensembles both revealed N-Myc binding to several parts of the AurA N-lobe and indicated extensive variability within the complex (Fig. 7a, b). To further investigate the degree of consistency of the SAXS-filtered ensemble with our independently obtained NMR and HDX-MS data, we first calculated the per-residue buried surface area both in the selected conformational ensemble and in all generated complex models (Supplementary Fig. 10c, d)[58]. The SAXS-filtered ensemble revealed a consistent subset of partially buried residues in both N-Myc and AurA, whereas the unfiltered ensemble revealed a broader set of buried residues. This suggests that the filtered ensemble describes a specific contact pattern despite the retained flexibility of the ensembles.

We continued to ask whether the SAXS-filtered ensemble could help identify potential residue-specific contacts between AurA and N-Myc, and therefore generated per-residue contact maps for both N-Myc$_{1-69}$–AurA and N-Myc$_{1-100}$–AurA complexes, and further calculated the fraction of contacts per residue in N-Myc and AurA across the

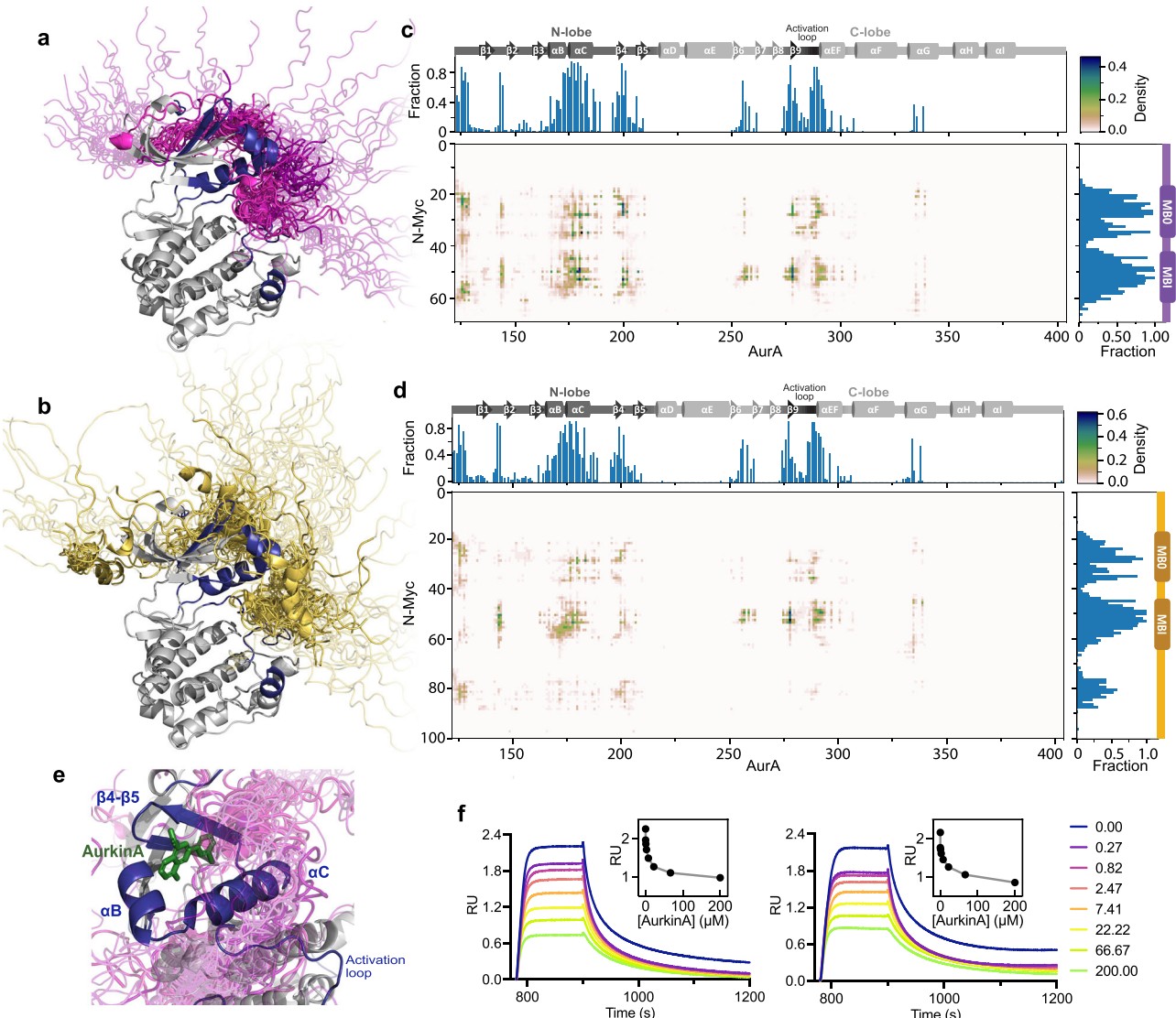

**Fig. 7 | Molecular modeling of the N-Myc–AurA complex reveals a targetable site near the regulatory spine of AurA.** Ensemble models of N-Myc$_{1-69}$ (purple) (**a**) and N-Myc$_{1-100}$ (gold) (**b**) in complex with AurA generated using AFsample. Structures include the 30 models with the best $\chi^2$ fit to the experimental SAXS data for the respective complex. AurA residues showing high degree of burial by N-Myc across the SAXS-filtered ensembles are marked in blue. Buried residues on N-Myc are shown as cartoon, while non-buried regions are shown in ribbon-representation. Residue-residue contact maps between N-Myc$_{1-69}$ (**c**) or N-Myc$_{1-100}$ (**d**) and AurA, colored by the density of the contact in the SAXS-filtered ensembles. On top and right of the contact maps, bar-graphs indicate the fraction of contact for each residue in AurA (top) and N-Myc (right), with the respective sequence or structure motifs indicated in line with the residue numbering. **e** Structure of AurkinA (green) binding AurA (PDBID: 5DT4) in the context of the AFsample ensemble for AurA binding N-Myc$_{1-69}$ (purple). **f** Bilayer interferometry (BLI) measurements for AurA binding to N-terminally immobilized N-Myc$_{1-69}$ and N-Myc$_{1-100}$ in the presence of increasing concentrations of the AurA inhibitor AurkinA.

selected ensembles (Fig. 7c, d). Partially protected residues were identified throughout MB0 and MBI in both N-Myc$_{1-69}$ and N-Myc$_{1-100}$, in line with our experimental NMR and biophysical data. Within the ensembles, and in agreement with HDX-MS results, several regions in the AurA N-lobe were significantly buried and showed a high fraction of contact in both complex ensembles. The highest degree of burial was observed within residues 171–189, which encompass the αC-helix and part of the β4-connecting loop. The AurA N-terminal residues 125–128, where peptides were not captured by HDX-MS, were also consistently protected throughout the ensembles. A smaller fraction of contacts were seen for residues 255–258, located within the protected αE-peptides in the HDX-MS experiments, and moderate burial was observed within the AurA activation loop (residues 270–293), where transient N-Myc–AurA interactions cannot be ruled out although this loop appeared highly dynamic both in the absence and presence of N-Myc in the HDX-MS experiment. Very few contacts were

identified for AurA residues 334-337 in the C-lobe αG helix, previously postulated as a major site for N-Myc binding[18], in agreement with the slight perturbations observed in the HDX-MS experiments (Fig. 5, Supplementary Fig. 6). The burial pattern on N-Myc agrees closely with the amino acid regions identified by NMR to be affected by AurA, both in N-Myc$_{1-69}$ and N-Myc$_{1-100}$. Reciprocally, the AurA N-lobe is partly buried in all models, in agreement with this being the main protected region observed by HDX-MS. Although only SAXS experimental data was used to confine the ensembles, their consistency with NMR and HDX-MS data independently confirms the validity of the computational approach.

### Blocking the AurA N-lobe Y-pocket inhibits N-Myc$_{1-69}$ and N-Myc$_{1-100}$ binding

Based on our finding that N-Myc interacts with the AurA N-lobe, mediated by MB0 and MBI, and thereby both stabilizes and activates

AurA, we asked whether this interaction could be specifically inhibited. TPX2 has been shown to compete with N-Myc binding to AurA[18], and we therefore asked if an inhibitor designed to block TPX2-AurA binding to the N-lobe could interfere with N-Myc binding. We chose the AurA inhibitor AurkinA, which impairs TPX2 by binding to the AurA Y-pocket between αB and αC in the AurA N-lobe (Fig. 7e), as our work shows that this AurA site is also protected by N-Myc (Fig. 5). In competition assays monitored by BLI, we found that AurkinA, which has comparable affinity for AurA to N-Myc ($K_D = 10.6 \,\mu M$)[59], was able to efficiently compete with both N-Myc$_{1-69}$ and N-Myc$_{1-100}$ for binding to AurA (Fig. 7f). These results reveal the potential for targeting the AurA N-lobe to therapeutically disrupt the N-Myc−AurA complex.

## Discussion

MYC proteins have often been referred to as inherently "un-druggable" due to their intrinsically disordered nature. An alternative approach to inhibit MYC activity may be to disrupt interactions with key partner proteins, but this requires in-depth understanding of the molecular and dynamic properties of the targeted interaction. For N-Myc driven cancers, the interaction with AurA has emerged as a promising therapeutical target, but limited success has been seen in clinical trials to date[32–34]. Recent works have built on the view that N-Myc interacts with AurA in a folding-on-binding manner, involving a small, non-conserved region of N-Myc that interacts with AurA in a helical conformation, presumably disrupting N-Myc degradation[18,36,37]. However, as multiple regions on N-Myc are known to be affected by AurA[18,35], this puts the current working model into question.

Our work reveals a functional N-Myc−AurA complex ensemble, determined using multiple independent yet complimentary experimental modalities. We show by a highly integrated biophysical and computational approach that the N-terminal region of N-Myc binds to the AurA N-lobe. Two binding regions in N-Myc$_{1-69}$, comprising aromatic patches within MB0 and MBI, contribute significantly to both binding affinity and functionality, as judged by the reduced binding and complex stability seen upon mutation of these sites and by their distinct ability to promote AurA kinase activity. A third binding region, similar to one identified earlier[18], is also clearly detectable by NMR and leads to slightly higher affinity in N-Myc$_{1-100}$ but does not alter the imprint on AurA as judged by HDX-MS, does not bind AurA on its own, and has very little impact on AurA activity.

The results presented here suggest an N-Myc−AurA complex with several binding modes that maintain complex formation despite low overall affinity. NMR and HDX-MS data identify binding to the AurA N-lobe by key N-Myc interaction regions in MB0 and MBI and are fully consistent with the SAXS-based conformational ensemble describing the N-Myc−AurA interaction. However, while the combined contact maps of N-Myc$_{1-69}$ and N-Myc$_{1-100}$ complex ensembles show a preference for N-lobe engagement, the interacting regions within N-Myc do not prefer discrete sites on AurA. Furthermore, and consistent with HDX-MS results, the complex ensembles reveal no additional AurA interaction site for N-Myc$_{1-100}$, despite its additional binding region. Although HDX-MS experiments show that AurA is significantly and similarly protected at several sites within the N-lobe by both N-Myc$_{1-69}$ and N-Myc$_{1-100}$ binding, N-Myc does not appear to fold in the N-Myc−AurA complex and retains high H/D exchange rates upon binding AurA. Moreover, the combined SEC-MALS and SEC-SAXS data support the formation of a monodisperse 1:1 complex, whereas NMR indicate rapid exchange and distinct dynamic profiles for the MB0 and MBI binding regions. Taken together, all these results hint at a mode of interaction where several regions in N-Myc contact multiple binding sites on AurA in a dynamic manner, resulting in a complex with multivalent and/or fuzzy properties[10,12].

Recent findings in the rapidly evolving IDP field suggest that their flexibility is key to responsive functional regulation in cells[10], where fuzzy interactions have evolved to be highly sensitive to modulating environmental conditions, often signaled by ligand binding and/or posttranslational modifications[60]. This concept is also highlighted by the multimodular modus operandi emerging for MYC regulatory interactions. For instance, Phosphatase1 nuclear targeting subunit (PNUTS) binding to c-Myc-MB0 increases access to the MBI phosphodegron region, thereby enhancing Protein Phosphatase 1 (PP1) access to dephosphorylate c-Myc-T58/S62, preventing c-Myc degradation[16,61]. This makes c-Myc susceptible to direct regulation via its protein-protein interactions, allowing its degradation to be tunable. Similarly, AurA binding to N-Myc$_{61-89}$ has been proposed to protect N-Myc from Fbwx7-mediated degradation[18,24]. In the context of N-Myc$_{1-100}$, we could not verify a significant and distinct N-Myc$_{61-89}$−AurA interaction in solution. However, we note that as the N-Myc−AurA crystal structure was obtained in the presence of ATP and using a pT58-N-Myc peptide that lacked part of the MB0 binding site, the alternative structural models could reflect consequences of contextual modulation, where multiple binding sites on N-Myc and AurA together with post-translational modifications could play a critical functional role in shifting the populations of alternative binding modalities. Crystal structures of Aurora regulatory binders TPX2[28], INCENP[62] and the inner centromere protein CEP192[63] show that they all bind the Aurora N-lobe by aromatic/aliphatic residues flanked by charged residues, as in N-Myc, but we could not identify distinct common sequence motifs. Further mapping of the structural and dynamic diversity in MYC complexes will therefore be required to fully understand the specificity of MYC functional modulation, and thereby how MYC interactions can be ultra-sensitive to context, ensuring efficient sensing of cellular signaling[11].

Extending the current understanding of kinase activation, our data suggests that a dynamic and possibly multistate interaction may be sufficient to modulate kinase activity. We show that N-Myc binding engages key elements that mediate allosteric communication in kinases, such as αB, αC, and the αC−β4 loop[27,52]. Activation of Aurora kinases by INCENP and TPX2 has been proposed to rotate αB and αC in a manner that engages the spine and active site region[28,62]. However, previous NMR studies suggest that the AurA active site region and spine comprise multiple states in intermediate exchange[29], and in-depth DEER investigations suggest that TPX2 binding to the AurA N-lobe activates the enzyme by allosterically remodeling the distribution of such states[64]. Similarly, recent studies of the Abl kinase have shown that the relative occupancy of conformational states in the DFG motif, αC and activation loop determine its enzymatic activity, and that these populations can be modulated by regulatory domains, inhibitors, or mutations[65]. Our results agree with the hypothesis that allosteric activation of kinases is dynamics-driven[27,65] and further extends its reach by suggesting a mode of action for how IDP interactions could allosterically activate a kinase such as AurA. In a wider context, kinase domains are often activated by N- and C-terminal tail regions with IDP-like characteristics and resolving how these regulate each protein kinase provides a fundamental challenge for the biological community[66].

Thorough analysis of how IDPs interact with their targets is increasingly being recognized as a route towards therapeutic opportunities. In this respect, our data strongly supports a broadening of efforts to target the N-Myc−AurA complex by considering inhibition of its interactions within the AurA N-lobe, as revealed here[67]. As our HDX-MS results reveal that the Y-pocket is highly protected by N-Myc MB0/MBI and is a critical part of the binding interface in the ensemble models, the accessibility of the Y-pocket is likely critical to the N-Myc−AurA interaction. In support of this, we show that the inhibitor AurkinA, designed to disrupt TPX2 binding by blocking the hydrophobic Y-pocket, also competes efficiently with N-Myc binding to AurA, providing exciting possibilities for AurkinA-like inhibitors such as its high-affinity analog CAM2602[68]. We propose that further therapeutic focus on targeting the AurA N-lobe would increase the

probability of obtaining an efficient N-Myc–AurA inhibitor which could be used as a therapeutic to disrupt this critical interaction in high-risk tumors.

## Methods

### Production of N-Myc proteins

Human N-Myc (residues 1–69 or 1–100) were subcloned into the pNH-Trxt vector (Addgene #26106). The protein was overexpressed in *E. coli* BL21(DE3) cells in LB medium supplemented with 50 µg/ml kanamycin by induction with 0.5 mM 1-thio-β-D-galactopyranoside (IPTG) at an $OD_{600}$ of 0.6 at 37 °C for 3 h. Harvested cells were resuspended in lysis buffer (50 mM $NaPO_4$, 10 mM Tris-HCl, 100 mM NaCl, 10 mM imidazole, 5% glycerol, 5 U/ml DNaseI (Roche), cOmplete EDTA free protease inhibitor (Roche), 10 mM β-mercaptoethanol, pH 8.0) and sonicated on ice. Following centrifugation, the supernatant was affinity purified at 4 °C using $Ni^{2+}$-NTA resin (Invitrogen™ R90110) and eluted with 300 mM Imidazole. Following dialysis into 40 mM NaCl, purification was done by anion exchange using a 5 ml XLHighTrap column (Cytiva) and eluted by gradual increase in NaCl to 400 mM. The 6x-His-Trx-tag was cleaved with TEV at 4 °C during dialysis overnight and removed by $Ni^{2+}$ chromatography using 300 mM NaCl. Final polishing was done using a HiLoad 16/600 Superdex 75 column (Cytiva) equilibrated in GF buffer (50 mM NaPO4, 10 mM Tris-HCl, 100 mM NaCl, 5% glycerol, 10 mM β-mercaptoethanol, pH 8.0).

For biotinylated proteins, human N-Myc fragments (residues 1–69 or 1–100) were subcloned into a modified pET15-MHL vector (Addgene #26092) such that they contained an N-terminal 6xHis-Thioredoxin tag cleavable by TEV protease, and a C-terminal AviTag (GLNDI-FEAQKIEWHE). Proteins were co-expressed with biotin ligase BirA from the pCDF-BirA vector (GenBank ID #JF914075) from *E. coli* BL21(DE3) cells in TB medium supplemented with 100 µg/ml ampicillin, 50 µg/ml spectinomycin, and 20 µM D-biotin. Protein expression was induced overnight at 16 °C with 0.5 mM IPTG when the cells reached $OD_{600} \approx 0.8$. Proteins were purified according to the non-biotinylated N-Myc protocol, excluding the anion exchange step.

### Peptide synthesis

N-Myc peptides corresponding to residues 61-90 and 69-90 of human N-Myc were synthesized using Fmoc-chemistry on an automated microwave peptide synthesizer (Liberty Blue, CEM, Matthews, North Carolina, USA) in 100 µM scale. ProTide Rinkamide (LL) resin was used as solid support for synthesis yielding an amidated C-terminal. For synthesis, Fmoc-protected amino acids were sequentially coupled using a four-fold excess of amino acid, Oxyma as base and DIC as coupling reagent in DMF under microwave conditions. Fmoc-deprotection was achieved by treatment with 20% Piperidine in DMF under microwave conditions. After final Fmoc-deprotection N-terminal acetylation was done by acetic anhydride and DIEA (5 eq) in DMF for 1 hr. Global deprotection and cleavage of peptides from resin was achieved by treatment with TFA:$H_2O$:Phenol:DTT (88/5/5/2, v/v/v/v) for 3 h before being concentrated using a stream of nitrogen. The crude peptides were precipitated in ice cold diethyl ether, twice, and the ether was discarded. The crude peptides were purified on a semi preparative HPLC system (Dionex) equipped with a RP C-18 column (ReproSil Gold) using a gradient of acetonitrile containing 0.1% TFA. Peptide identity and purity was confirmed using MALDI-ToF mass spectrometer (Bruker) and HPLC (Thermofischer) respectively (Supplementary Fig. 11).

### Production of Aurora A protein kinase

Aurora A kinase residues 122-403, with the double Cys-mutations C290A:C393A described previously[38] and with or without the phosphorylation mimic mutation T287A:T288E, was subcloned into two different vectors: pNH-Trxt (Addgene #26106), which were transformed into *E. coli* BL21(DE3) cells containing the ROS-2 pRAR3 plasmid, and SUMO-6xHis-Lic1.10 (a gift from Patrick Celie at Netherland's Cancer Institute[69]), which was transformed into *E.coli* BL21(DE3) pLysS cells (Sigma Aldrich). Protein overexpression was done in LB medium supplemented with 50 µg/ml kanamycin and 34 µg/ml Chloramphenicol at 18 °C overnight following induction with 0.5 mM IPTG at an $OD_{600}$ of 0.6. Harvested cells were resuspended in lysis buffer (50 mM Tris-HCl pH 7.5, 300 mM NaCl, 5 mM $MgCl_2$, 10 mM imidazole, 10% glycerol, 5 U/ml DNaseI (Roche), cOmplete EDTA free protease inhibitor (Roche), 10 mM β-mercaptoethanol) and sonicated on ice. The protein was affinity purified at 4 °C using Ni-NTA resin (Invitrogen™ R90110) following centrifugation. The 6x-His and solubility tag was cleaved with SENP2 during dialysis at 4 °C overnight and the protein was purified through incubation with Ni-NTA resin before the final polishing step by a HiLoad 16/600 Superdex 75 column (Cytiva) using GF buffer (50 mM Tris-HCl pH 7.5, 150 mM NaCl, 10% glycerol and 10 mM β-mercaptoethanol). Unphosphorylated $AurA_{122-403}$ with mutations C290A:C393A was produced by co-expression with λ-phosphatase[54] (a gift from John Chodera, Nicholas Levinson and Markus Seeliger (Addgene plasmid # 79748; http://n2t.net/addgene: 79748; RRID: Addgene_79748) in BL21(DE3) cells. The co-transformation was carried on in a consecutive manner, where the λ-phosphatase was transformed first to BL21(DE3) and once the transformation was confirmed, it was followed by the transformation of AurA. The co-expressed unphosphorylated $AurA_{122-403}$ C290A:C393A was purified as mentioned above.

### Size-exclusion chromatography with multi-angle light scattering (SEC-MALS)

SEC-MALS analysis was done by injecting samples onto a Superdex 75 Increase 10/300 column (Cytiva) equilibrated in the sample buffer (50 mM NaPO4 (pH 8), 10 mM Tris, 10 mM BME, 5% v/v glycerol, 100 mM NaCl) with a flowrate of 0.5 mL/min. Complex formation was done prior to injection by mixing N-Myc constructs with AurA in a 2:1 molar ratio followed by incubation for 2 h on ice. The MALLS-RI data were measured using Mini-Dawn TREOS module coupled to an Opti-Lab T-Rex refractometer and a DynaPro Nanostar (Wyatt Technologies). The molecular weight distribution of species eluting from the SEC column was evaluated using ASTRA 6.1 (Wyatt Technologies).

### Dynamic light scattering (DLS)

DLS of N-Myc$_{1-69}$ and N-Myc$_{1-100}$ was measured on a 100 µM sample in the sample buffer (50 mM $NaPO_4$, 10 mM Tris-HCl, 100 mM NaCl, 5 %v/v Glycerol, 10 mM b-mercaptoethanol, pH8.0), using a DynaPro Plate reader (Wyatt Technologies) at 20 °C using ten acquisitions at 5 s. Data was analyzed using the accompanied software DYNAMIC 7.10.1 (Wyatt Technologies).

### Isothermal titration calorimetry (ITC)

ITC measurements were performed on a Microcal PEAQ-ITC (Malvern) at 25 °C, using a reference power of 10.0 µcal/s and a stir speed of 750 rpm. Samples were dialyzed into the ITC buffer (20 mM Hepes pH 7.5, 150 mM NaCl, 5 mM $MgCl_2$, 5 %v/v glycerol, 2 mM TCEP), and assayed using N-Myc constructs at approximately 400 mM, which were titrated into a 45 mM AurA sample over 19 injections of 2 µl each. Data analysis was done using using a 1:1 binding model in the MicroCal PEAQ-ITC Analysis Software (Malvern), where the baseline was manually adjusted, and where the 1:1 binding model was validated by stochiometry assayed by SAXS and SEC-MALS. As more complex binding models did not improve the fit to data for the strongest binders, and to allow direct comparisons between constructs with a different number of binding regions, we applied the 1:1 ITC model to all data.

### Small-angle X-ray scattering (SAXS)

SAXS data collection was performed on the P12 bioSAXS beamline[70] at PETRA III (Hamburg, Germany) using in-line SEC-SAXS setup for AurA,

N-Myc$_{1-69}$, N-Myc$_{1-100}$, and their respective complexes. Parallel MALLS-RI measurements[71] were recorded for AurA, N-Myc$_{1-100}$, and the respective complex. The protein samples were injected onto Superdex 75 Increase 10/300 equilibrated in the sample buffer (20 mM HEPES 150 mM NaCl 5 mM MgCl$_2$ 3% v/v glycerol 2 mM TCEP pH 7.5) with a flowrate of 0.70 mL/min. The 2400 individual frames for each run were processed using SASFLOW[72] and CHROMIXS[73] to obtain final buffer-subtracted scattering profiles, and the data analysis was performed using the ATSAS 3.2.1 suite[74]. The radius of gyration, R$_g$, and forward scattering intensity, $I$(0), were obtained using Guinier approximation[75] and GNOM program[76] was used to evaluate a paired distance distribution function $p(r)$ and to obtain maximum particle dimension, D$_{max}$. Molecular weight estimates were calculated using concentration-independent methods including DatBayes[77], volume of correlation, V$_c$[78], SAXSMoW[79]. Estimation of $\Delta\rho$ was done using MULCh[80]. The MALLS-RI data were measured using Mini-Dawn TREOS module coupled to an OptiLab T-Rex refractometer (Wyatt Technologies). The molecular weight distribution of species eluting from the SEC column was evaluated using ASTRA7 (Wyatt Technologies). The detailed information about samples, data collection and processing, and structural parameter analysis are outlined in Supplementary Table 1, in accordance with the publication guideline[81]. Data and models are deposited in SASBDB[82].

## NMR spectroscopy (NMR)

Isotopically labeled N-Myc (residues 1–69 and 1–100) was over-expressed in BL21(DE3) cells in M9 minimal media (6 g/l NaHPO$_4$, 3 g/L KH$_2$PO$_4$, 0.5 g/L NaCl, 50 μM CaCl$_2$, 1 mM MgSO$_4$) supplemented with 10 ml/L Gibco Vitamin solution, 2 g/L $^{13}$C-based glucose, 1.5 g/L $^{15}$NH$_4$Cl and 50 μg/ml kanamycin. Expression was induced at an OD$_{600}$ of 0.6 using 0.5 mM IPTG and incubation at 37 °C for 3 h. Purification was done as described for unlabeled sample above. Following buffer optimization using DLS and SEC-MALS, NMR experiments were performed in the optimized buffer (20 mM MES (pH 6.5), 100 mM NaCl, 5% v/v glycerol, 5 mM DTT, 10% D$_2$O, 100 μM NaN$_3$, 1 mM TCEP). Assignment of N-Myc$_{1-69}$ and N-Myc$_{1-100}$ was done at 164 μM and 98 μM, respectively, using a series of vnmrj BioPack resonance experiments ($^1$H$^{15}$N-HSQC, CBCAcoNH, HNcaCO, HNCO, HBHAcoNH) and a HNCACB sequence retrieved from the Kay lab, recorded at 288 K on a Varian IN-OVA spectrometer (operating at 600 MHz) equipped with a cryogenic probe. All triple resonance experiments were nonuniformly sparsely sampled to 15% and all spectra were processed using NMRPipe[83] and MddNMR[84] and visualized and manually assigned using the software NMRFAM-Sparky[85]. For binding assays with AurA, unlabeled AurA was titrated to $^{13}$C$^{15}$N-labled N-Myc with a concentration of 240 μM (1-69) or 98 μM (1–100). $^1$H$^{15}$N-HSQC were recorded at molar ratios of 0, 7, 12, 21, and 28% AurA for N-Myc$_{1-69}$ and 0, 7, 12, 16, 23, and 30% AurA for N-Myc$_{1-100}$. I/I$_0$ was calculated following normalization based on N-Myc concentration. CSPs were calculated as previously described[20].

The $^{15}$N longitudinal relaxation rates ($R_1$), rotating frame relaxation rates ($R_{1\rho}$), and {$^1$H}-$^{15}$N heteronuclear Overhauser effects (hetNOE) were measured on 250 μM N-Myc$_{1-69}$ samples in the same buffer as above, at 288 K on a Bruker Avance Neo (operating at 600 MHz) equipped with a TCI cold probe, using Bruker HSQC-detected pulse sequences: hsqct1etf3gpsitc3d ($R_1$), hsqcretf3gpsitc3d ($R_{1\rho}$), and hsqcnoef3gpsi3d (hetNOE), running on Topspin 4.5.0. Experiments were performed for $^{15}$N-labeled N-Myc$_{1-69}$ alone and in substoichiometric concentrations of AurA (6 μM, 15 μM and 25 μM). $R_1$ relaxation delays (ms) were 20, 40, 60 (triplicate), 90, 120, 200 (duplicate), 400 (duplicate), 600 and 800 ms. $R_{1\rho}$ relaxation delays (ms) were 4, 24 (duplicate), 48, 72, 96 (duplicate), 140 (duplicate), 170 and 200 ms. hetNOE spectra with and without proton saturation were acquired in duplicates. Analysis of all relaxation experiments was done using PINT[86,87], including conversion of $R_{1\rho}$ to transverse relaxation rates ($R_2$)

using $R_1$. Error bars were calculated based on the repeats indicated above using the jackknife method embedded within PINT. The $^1$H$^{15}$N-HSQC of N-Myc$_{1-69}$ in the absence (1:0) and presence (1:1) of AurA (50 μM) were recorded under the same experimental conditions as the relaxation data, except for the protein concentrations. Spectra were collected using a variant of hsqcetfpf3gpsi2 modified by Maxim Mayzel, obtained from the Swedish NMR center (Gothenburg, Sweden).

## Hydrogen-deuterium exchange mass spectrometry (HDX-MS)

HDX-MS was performed on three separate occasions to investigate 1) N-Myc$_{1-100}$ binding to AurA, 2) N-Myc$_{1-69}$ binding to AurA, and 3) AurA binding to N-Myc$_{1-100}$ and the reverse. In run 1 and 2, 45 μM AurA was mixed 1:1 v/v with 120 μM N-Myc$_{1-100}$, 180 μM N-Myc$_{1-69}$ or TBS (50 mM Tris-HCl, 150 mM NaCl, pH 7.5), yielding apo and bound AurA for comparison. For reciprocal analysis 60 μM N-Myc$_{1-100}$ and 60 μM AurA were mixed 1:1 v/v, and apo controls were prepared similarly with TBS, making it possible to qualitatively probe the protection of both proteins in the same complex and confirm consistency to previous runs. All mixtures were incubated on ice for 1 h before further handling.

HDX-MS experiments were run in triplicates using automated sample preparation on a LEAP H/D-X PAL™ platform (Trajan Scientific and medical) coupled to an Ultimate 3000 micro-LC and an Orbitrap Q Exactive Plus MS (Thermo Scientific). All reagents were from Sigma Aldrich. A 4-point calibration (pH 2, 4, 7, 10) was made prior to all measurements using a SevenCompact pH-meter with an InLab Microelectrode (Mettler-Toledo). Samples (3 μl) were diluted 10-fold with TBS (50 mM Tris-HCl, 150 mM NaCl) or a D$_2$O buffer of the same composition (pH$_{(read)}$ 7.5 and 7.1). Exchange was carried out for t = 0 s, 30 s, 300 s, 3000 s, and 9000 s at 4 °C. The labeling reaction was quenched by a 1:1 dilution (1% TFA, 0.4 M TCEP, 4 M urea, pH 2.5) at 1 °C. Quenched samples (55 μl) were directly injected and subjected to online digestion at 4 °C at 50 μL/min for 4 min (0.1 % formic acid (FA), pH 2.5) using an in-house immobilized 2.1 × 30 mm pepsin column (run 1-2) or a 2.1 × 20 mm Nepenthesin-2/Pepsin mixed digestion column (AffiPro, CZ) (run3, for increased sequence overlap/resolution of the relatively small N-Myc). Peptides were subjected to on-line SPE on a PepMap300 C18 trap column (1 mm x 15 mm) and washed with 0.1% FA for 60 s. The trap column was switched in-line with a C18 reversed-phase analytical column (Hypersil GOLD, particle size 1.9 μm, 1 ×50 mm) for separation at 1 °C (8 min 5–50 % B (95 % acetonitrile/0.1 % FA), then 50–90% B for 5 minutes). Carry-over between injections were minimized by equilibrating the trap and column at 5% organic content, cleaning the needle port and sample loop three times after each injection (5% MeOH/0.1% FA, followed by 90% MeOH/0.1% FA and finally 5% MeOH/0.1% FA), and washing the Pepsin column by injecting 90 μl of 1% FA /4 M urea /5% MeOH. A full blank was run between each sample injection. Separated peptides were analyzed on a Q Exactive Plus MS equipped with a HESI source operated at a capillary temperature of 250 °C with sheath gas 12, Aux gas 2, and sweep gas 1 (au). MS full scan spectra were acquired at 70 K resolution, AGC 3e6, Max IT 200 ms, and scan range 300-2000.

For peptide identification, undeuterated samples were analyzed using data-dependent MS/MS with HCD fragmentation in PEAKS Studio X (Bioinformatics Solutions Inc, BSI, Waterloo, Canada) and searched against the protein sequence FASTA file (15 ppm mass error tolerance, 0.05 Da fragment mass error tolerance, allowing for fully unspecific pepsin cleavage). Peptides with a peptide score value of -log$_{10}$ (P) > 25 were retained for HDX analysis in HDExaminer 3.1.1 (Sierra Analytics Inc, Modesto, US). Initial constraints disregarding modified peptides resulted in partial low coverage of the AurA sequence. Allowing for phosphorylation in the PEAKS peptide identification resulted in phosphorylated peptides across AurA residues 277-294 (data is available in PXD062561). As no significant difference in uptake was observed for phosphorylated and unmodified peptides,

phosphorylated T288 (as supported by literature[28]) was added to the evaluation to increased HDX-MS sequence coverage.

HDX analysis was made on charge states 1–6, allowing only for EX2, and assuming the first two residues of a peptide were unable to hold deuteration. Full deuteration was set to 75% of the maximum theoretical uptake in HDExaminer to compensate for the systemwide back-exchange and best reflect the uptake dynamics without having run a fully deuturated sample. This affects the %D scaling and heatmap coloring, but does not alter the reported number of deuterons. The presented deuteration data represent averages of high/medium confidence peptides within a $\pm 0.5$ min RT window; outliers and inconsistent peptides were removed after manual inspection. In the HDX protein-protein interaction analysis, bound vs apo states were compared within each run. When comparing N-Myc$_{1-69}$ and N-Myc$_{1-100}$ binding on AurA, only AurA peptides which were consistent between runs 1 and 2 were used, resulting in a common peptide pool of 96 representative AurA peptides for a robust and direct comparison between the experiments. Heatmaps were generated using heavy smoothing and uncolored prolines. Automatically calculated significance based on volcano plot significance lines were used for generating the difference heatmaps. A summary of the HDX-MS experimental detail is reported in Supplementary Table 3. The mass spectrometry and HDExaminer analysis files have been deposited to the ProteomeXchange Consortium via the PRIDE[88] partner repository.

### Biolayer interferometry (BLI)
BLI was performed using an Octet RED384 system (Forte Bio) with 150 nM biotinylated N-Myc$_{1-69}$ or N-Myc$_{1-100}$ immobilized on streptavidin (SA) biosensors (Sartorius). Binding assays were performed in the optimized buffer (20 mM Hepes pH 7.5, 150 mM NaCl, 5 mM MgCl$_2$, 5% v/v glycerol, 2 mM TCEP, 0.05% Tween-20 and 0.2 mg/ml BSA) using Octet 384 tilted well microplates (Sartorius) with AurA concentrations ranging from 0-30 μM. For the competition assay, 1 μM AurA was incubated with 0-200 μM AurkinA, and a final BSA concentration of 1 mg/ml.

### Nano differential scanning fluorimetry (nanoDSF)
NanoDSF measurements were performed using the Prometheus NT.48 (NanoTemper Technologies) and Prometheus Standard capillaries in the optimized buffer (20 mM Hepes pH 7.5, 150 mM NaCl, 5 mM MgCl, 2, 5% v/v glycerol, 2 mM TCEP). Complex formation was done by mixing AurA at a final concentration of 20 μM, with various N-Myc variants at final concentrations of 10, 20, 40, 100 and 200 μM. Thermal unfolding was measured by monitoring the 350 nm/330 nm fluorescence ratio over a temperature gradient of 25 °C to 80 °C with an increase of 0.3 °C/minute and visualized as the first derivative of the 350 nm/330 nm curve. Measurements were done on duplicate samples, except for N-Myc alone which did not show thermal melting. The onset and thermal melting points were determined using the instrument software PR.ThermControl 2.3.1 (NanoTemper Technologies).

### Aurora A kinase activity assay
Kinase activity of our recombinantly produced AurA was determined using the ADP-Glo™ Kinase Assay (#V6930) together with the Aurora A Kinase Enzyme System (#V1931) from Promega Corporation (Madison, USA). The ADP-Glo™ Kinase Assay was preformed using the substrate and reaction buffer included in the Aurora A kinase system, together with our recombinantly produced proteins, to quantify the amount of ATP consumed in the kinase reaction using luminescence detection.

Briefly, the kinase reaction was performed using AurA alone or in complex with various N-Myc constructs. Complex formation was done by incubating a mixture of 3 μM AurA and 24 μM N-Myc on ice. AurA apo samples had a concentration of 3 μM. Samples were split into triplicates of 3 μl in a white 384-well plate (Corning #3824), whereafter the samples were mixed with 2 μl ATP/substrate mixture, giving final concentrations of 150 μM and 0.1 mg/ml for ATP and the substrate

(MBP), respectively. The mixture was left for 40 min at room temperature to allow the kinase reaction to take place. Following this, ATP Depletion was done by addition of 5 μl ADP-Glo™ Reagent, after which ADP was converted to ATP using the luciferase/luciferin reaction, which generates the detectible luminescence, by addition of 10 μl Kinase Detection Reagent. To determine the ATP-to-ADP conversion in the kinase reactions, an ATP/ADP standard curve was generated by mixing different amounts of ATP and ADP and recording the luminescence for each sample. The assay was performed in a white 384-well plate (Corning #3824) on a CLARIOstar Microplate reader (BMG Labtech) using triplicates. Results are reported as % activation, calculated from the activity for AurA in complex with N-Myc compared to the activity for AurA alone.

### Molecular modeling using AFsample
To explore the conformational space comprehensively and increase the likelihood of identifying accurate structural models, we generated 10,000 structural models using the AFsample algorithm[56] without incorporating any templates. This large number of models allows for extensive sampling of possible conformations and enhances the probability of capturing the native-like structure of the proteins. To assess the fit of the generated models to the Small-Angle X-ray Scattering (SAXS) data, we employed Pepsi-SAXS[57]. This facilitated the evaluation of how well each model conformed to the experimental SAXS data ($\chi^2$) and computed Radius of gyration (Rg) for each model. The buried surface area formed by the interaction between N-Myc and AurA was calculated using Naccess[58]. This calculation involved the following steps: the solvent-accessible surface area (SASA) of each individual subunit (N-Myc and AurA) was computed separately, the SASA of the N-Myc-AurA complex was determined, and the difference in SASA between the individual subunits and the complex was calculated for each residue to ascertain the buried surface area. The total absolute buried surface areas were reported in square angstroms (Å$^2$). To investigate specific contacts between N-Myc and AurA, a residue-residue contact map was constructed, reflecting the frequency of contact between specific residues in N-Myc and AurA within a subset of the ensemble which had a $\chi^2<1.1$ to the experimental SAXS scattering profile for the respective complex. The ranking confidence was calculated as 0.8ipTM + 0.2pTM, where ipTM is the pTM for the interface, and pTM is the individual chains. To visualize the generated ensembles, the 31 top ranking models for N-Myc$_{1-69}$ and N-Myc$_{1-100}$, respectively, were aligned in PyMol based on the AurA structure (Supplementary Data 6-7).

### Reporting summary
Further information on research design is available in the Nature Portfolio Reporting Summary linked to this article.

## Data availability
All data needed to evaluate the conclusions in this paper are available in the manuscript and/or the Supplementary Materials and/or publicly available repositories. The SAXS data generated in this study has been deposited in the SASBDB database under accession codes SASDXU7 (N-Myc$_{1-69}$), SASDXV7 (N-Myc$_{1-100}$), SASDXW7 (AurA), SASDXX7 (AurA in complex with N-Myc$_{1-69}$) and SASDXY7 (AurA in complex with N-Myc$_{1-100}$). All HDX-MS data presented in this study has been deposited to the ProteomeXchange Consortium via the PRIDE partner repository under project identifier PXD062561. Peptide uptake curves are also provided as Supplementary Data 1-5 with this paper. NMR assignments have been deposited to the BMRB database under accession codes 53483 (N-Myc$_{1-69}$) and 53482 (N-Myc$_{1-100}$). All structural models generated in this study using AFsample, as well as related results obtained from Pepsi-SAXS and Naccess, have been deposited to Zenodo (https://doi.org/10.5281/zenodo.18376012). The 31 highest-ranking models for N-Myc$_{1-69}$ and N-Myc$_{1-100}$, respectively, are also provided as PyMol sessions with this paper (Supplementary Data 6-7).

All additional data generated in this study are provided in the Source Data file. Source data are provided with this paper.

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

## Acknowledgements

The authors gratefully acknowledge funding from the Swedish Research Council (2018-04390, 2018-04392; MS, AA, 2020-03352, 2024-05619; BW), The Swedish Cancer Society (20 1276 PjF 01 H, 23 3158 Pj 01 H; MS), and The Swedish Childhood Cancer Fund (PR2022-0107, PR2019-0143; MS, AA), Canadian Institutes of Health Research (FRN# 156167; LZP, FRN# 191891; CHA), HORIZON-2020 infrastructure MOSBRI (101004806 -

 

MOSBRI - H2020-INFRAIA-2018-2020/H2020-INFRAIA-2020-1; ET, TMGK, MS), the Swedish Foundation for Strategic Research (SSF) within the Swedish national graduate school in neutron scattering SwedNess (GSn15-00 08; MS, ZP), and the European Research Council (101044665 PROTECT; LPK, DA). The Structural Genomics Consortium is a registered charity (no: 1097737) that receives funds from Bayer AG, Boehringer Ingelheim, Bristol Myers Squibb, Genentech, Genome Canada through Ontario Genomics Institute [OGI-196], EU/EFPIA/OICR/McGill/KTH/Diamond Innovative Medicines Initiative 2 Joint Undertaking [EUbOPEN grant 875510], Janssen, Merck KGaA (aka EMD in Canada and US), Pfizer and Takeda. We acknowledge LiU support from the Life Science Technologies Profile Area, and the ProLinC core facility for instrument access. Support from the Swedish National Infrastructure for Biological Mass Spectrometry (BioMS) and the SciLifeLab Integrated Structural Biology platform is gratefully acknowledged, and Lucas Hultgren for experimental assistance. The computations were performed on resources provided by LiU and NSC. We thank EMBL Hamburg for access to the P12 bioSAXS beamline and Dr. Melissa Graewert and Dr. Cy Jeffries for their assistance at the beamline and in data evaluation. We acknowledge SwedNMR nodes at Linköping and Göteborg University for infrastructure access and Dr. Ulrika Brath for experimental support, Dr. Isak Johansson-Åkhe, Matilda Sjöbom and Linnea Pierre for their contributions in early stages of this project, and Drs. Lars-Göran Mårtensson and Katherine Stott for helpful discussions.

## Author contributions

J.H., V.M., D.R., L.Z.P., A.A., and M.S. conceptualized the study, and J.H., V.M., A.A., and M.S. designed the experimental strategy. J.H. and V.M. produced the proteins. S.E. and J.H. performed and evaluated the HDX-MS. L.P.K. and D.A. designed and synthesized the N-Myc peptides. E.T., J.H., and V.M. produced the N-Myc mutants and performed the ITC and nanoDSF together with D.D. J.H. and A.A. performed the NMR experiments and analyzed the data. J.H. and Z.P. performed the SAXS experiments and analysis. J.H. and B.W. performed and evaluated the computational modeling. J.H. performed the DLS and SEC-MALS advised by D.D. J.H. performed the kinase activity assay. T.M.G.K. produced biotinylated N-Myc and performed BLI measurements jointly with J.H., supervised by C.H.A. and L.Z.P. J.H. and M.S. wrote the first draft and J.H. prepared the figures, with input from all co-authors. M.S. was the principal investigator on the project, set up the collaborations, and supervised and co-evaluated all results in all stages of the investigation.

## Funding

## Competing interests

MS is a co-founder and shareholder of MyCural Therapeutics, a company which develops small-molecules targeting the MYC-MAX interaction. No other author declare any competing interests.
