## [Transparent Peer Review file · Nature Communications]

The N-Myc MB0-MBI region interacts specifically and dynamically with the N-lobe of Aurora kinase A

Corresponding Author: Professor Maria Sunnerhagen

Version 0:

Reviewer comments:

Reviewer #1

(Remarks to the Author)

In this manuscript, Hultman et al perform an integrative study of the N-Myc – Aur interaction, providing insights into the architecture of this complex and the molecular determinants for complexation. The manuscript is generally well constructed, and the set of experiments performed is logical. However, in places there is a lack of detail which may hamper a non-expert from being able to understand the data presented. My area of expertise lies in the application of HDX-MS and other MS tools for structural applications, and as a result my comments are primarily aligned with this portion of the work.

- The authors perform extensive SEC-MALS and SEC-SAXS analysis. They present the resultant data in various ways – e.g. Guinier analysis(line 154) and p(r) analysis (line 156). It would be helpful for the authors to provide further detail as to how the data presented support their conclusions. For example, this could be included in the appropriate figure legends. As written, for a non expert in these techniques it is hard to understand how the authors have drawn the conclusions presented.
- Why are the authors showing two runs of AurA data in Figure 4C? This is confusing to the reader and is not discussed in the text/Methods. They are pretty well identical and im not sure both are needed to be shown here.
- More labels are needed in Figure 4. For example it is not clear to me the colouring shown in the top part of panel A. The authors state that the colouring here represents the innate HDX pattern, but that is a vague statement and further details are needed. I assume this has to do with percentage uptake from maximal deuteration but the 'key' on the right of the figure should be labelled to indicate this. The authors also need to state why they have chosen a '75 %' uptake (from Methods) to mean 100 % uptake for such an analysis. This is non-standard in the field from my experience.
- Label scale bars shown in lower part of 4A and Figure 4B. Might also be useful to add labels to the structures shown in Figure 4D given that they refer to key regions in the structure in the text.
- The authors state different residues are involved in interactions – e.g. 162-184 (line 239). But the data where protection is referred to doesn't directly correlate with these exact residue numbers (at least in terms of the labels shown). How were these numbers selected?
- How do the authors define 'strong' and 'weak' HDX protection (e.g. line 240 and Figure 4D)? This needs clarification and a metric applied. Was any statistical analysis used to identify protected regions?
- Given the difference in affinities, which the authors measure here, what is the difference in percentage bound during the HDX experiments for the two Myc constructs studied? This could be calculated and be helpful in interpreting the data showed. This should be stated in the text and a discussion of this included.
- The authors display an output from PEAKS in Supplementary Figure 4 where putative phosphorylation sites are indicated. In my experience, PEAKS can annotate very low confidence PTM sites in these types of reports, where the MS/MS data do not provide the requisite support for the presence of PTMs at a given residue. The authors should investigate their data further, and include MS/MS spectra for all phosphosites that are confidently identified. I would recommend changing this figure to one that only shows the localisation of confident phosphosites after this manual data curation has been performed.
- In the methods indicate the protein concentrations used – this is critical for understanding the percentage bound in these experiments.
- What do the authors mean by the FASTA file only containing the RDB sequence (line 703)?
- The authors must indicate the PRIDE dataset identifier in their manuscript – at the moment this is missing (line 699)
- The authors must include experimental details for their phosphosite analysis.

Minor comments:

Line 121 – add 'mass' before spectrometry.

In several places references are missing. E.g. line 478 and line 715. Please thoroughly proof-read the manuscript.

Reviewer #2

(Remarks to the Author)

The N-Myc MB0-MBI region forms a specific, functional and fuzzy complex with the N-lobe of Aurora kinase A, Hultman et al.

The manuscript describes the interaction between N-Myc and Aurora A. Disordered protein regions, like those found in Myc, are increasingly being viewed as potential drug targets. Given its near ubiquitous role in many cancers, Myc is frequently cited as a potential target. Finding interfaces that are formed when an IDR binds to another protein are an attractive alternative to simply trying to identify small molecules that can bind to IDRs in isolation. The work presented here not only does a great job in characterizing the specific interaction between Myc and Aurora A, but also lays out a straight-forward (albeit using a number of biophysical techniques) approach for characterizing such interfaces in general. The manuscript is written in a clear and succinct fashion and is easy to read and follow. The figures are all well organized and clear (I particularly appreciated the unified color things throughout the entire paper).

The authors use a combination of multiple, complementary, biophysical techniques that both report on structural features and affinities. Given what is required to obtain good quality NMR spectra on a protein like Aurora, I appreciated the use of the HDX as a more cost-effective and simpler (albeit lower resolution) alternative. With some revisions, this manuscript is certainly worthy of publication.

1. The use of “fuzzy” and “dynamic” to describe the complex, while quite possibly true, is not heavily supported in the data presented. These terms explicitly describe a dynamic process, and the vast majority of data in the paper is ensemble averaged and does not directly report on dynamics. The one exception is the HDX. I think the interpretation of the time scales of protection do hint at some dynamics in the complex, but the long time point (9000 s) is really quite long and I’m not sure how these exchange dynamics would compare to any complex, dynamic or no, with micromolar affinity. The authors also cite other NMR studies of IDPs (such as Milles et al) to suggest that their NMR data is in agreement with other studies of dynamic complexes. What is missing in this comparison is the careful NMR relaxation dynamics experiments that are present in those papers. If the authors were to include R1 R2 hetNOE and relaxation dispersion analysis, the point would be significantly stronger. This data would also significantly enhance the paper’s conclusions.
2. The authors conclude that the significant line broadening on addition of Aurora is the result of intermediate exchange (and thus also indicative of a dynamic complex). While it is possible that this is true, a more likely explanation is simply that residues in complex with Aurora are tumbling with a much different τ_c and are broadened relative to the other disordered residues. This is quite common in complexes of IDRs with larger, folded, binding partners regardless of the exchange regime. In order to make this point, again, a much more detailed accounting of the NMR relaxation should be done.
3. The conclusion that the complex is 1:1 is based on SEC-MAL measurements where the starting stoichiometry was only 2:1. It isn’t clear to me that either 1. There isn’t evidence of a higher MW complex at the front edge of the complex elution peak or 2. That there couldn’t be a higher molecular weight complex where a second MYC (or second kinase) is added with a much lower affinity. For example, this paper from Ben Schuler’s group describes an archetypal dynamic complex: <https://www.nature.com/articles/nature25762>. The original work concluded it was a 1:1 complex. However, later investigation at different concentrations revealed the existence of trimers and even phase separated droplets: <https://www.nature.com/articles/s41586-023-06329-5>. I think that it is likely fair to conclude that a 1:1 complex largely dominates the measurements shown here, but I don’t know if its reasonable to conclude that a 1:1 complex is the only complex that can form. I agree, with the authors that the SAXS data as shown also suggests a 1:1 complex. However, we do not see what the elution profiles look like for these data and the same concentration dependence would apply here as does for the SEC-MALS.
4. The nano DSF data is very compelling. It is very clear that it correlates with the HDX and show specific residues that are critical to the interactions. It is also great to see that this correlates with activity. One minor comment is that this should be referred to as “DSF” data and not “Prometheus” data in the text.
5. I liked the structural modeling section. Perhaps I have missed this in the literature, but, to my knowledge, this is the first use of AF structures in this way. That said, I think the more appropriate way to have approached this (as opposed to sorting by how well individual structures fit the experimental data) would be to use one of the methods that is established in the IDP field for fitting saxs data to ensembles. Either the EOM method included in the ATSAS package or any type of ensemble reweighting (<https://pubs.acs.org/doi/10.1021/acs.jctc.8b01231> for example). Given that the structural details seemed quite similar for the full ensembles versus the filtered, I don’t suspect there would be a huge difference in the conclusions for this paper, but the method would be more appropriate and the final results could be a bit cleaner.
6. Finally, I think that it is worth noting that the introduction contained a great level of detail that keys the reader in to why this particular interaction could be important in a broader biological setting. This is not always the case in biophysics papers.

A few small details

“most human cancers” in the introduction – I would prefer most specifics

Line 121 “several additional techniques” – there aren’t many more to list, perhaps just list them

145 – MYC is an IDR based on DLS? Also, based on the long retention time on the SEC column, presumably the SAXS profile and the NMR HSQC peak dispersion

163 – SAXS Kratky analysis is referenced, but is not explained

167 – It is worth noting that the two MYC constructs overlay nearly perfectly, also supporting the fact that they are disordered and there are no significant long-range interactions

478 Missing ref?

Reviewer #3

(Remarks to the Author)

The manuscript by Hultman and coworkers show an integrative structural biology approach to describe the complex between the first 100 residues in the transcription factor N-Myc and the Aurora A kinase. They conclude that the complex is “fuzzy” at 1:1 stoichiometry and formed with a strength of 1-4 micromolar. They show that binding to Aurora A leads to its activation and that binding can be inhibited by a competitive binding partner. The data have been carefully collected, and the manuscript is well written and provide new insight into the system under investigation. However, I have some reservations about the conclusion they reach - that the data supports a complex of multiple exchanging bound states; a conclusion – if true, that will need much more data to be confirmed, as detailed below. Also, there are already some studies on dynamic IDP – kinase interactions that have not been mentioned and referred to, but which are highly relevant for this study.

My main reservation is related to the statement of this complex being dynamic and fuzzy. This conclusion is reached from disappearance of the peaks from the NMR spectrum and lack of protection against exchange monitored by HDX-MS. This does not necessarily mean that the complex is disordered. The problem is that we cannot observe the bound state (see comments to HDX-MS below). First, there are no experimental data on the dynamics (with is provided by rates) of this complex, so how can the authors interpreted it as dynamic? Not forming secondary structure in binding can be a case where no protection will emerge in HDX MS. The disappearance of the NMR peaks may simply be due to exchange broadening between two states – a free and a (single) bound state – caused by the exchange rate being in the intermediate exchange regime for NMR and not due to the presence of multiple states in the bound state (fuzzy). The conclusion that these regions “shows extensive dynamics” are not supported by the data.

It will be necessary to record data on the bound state to provide insight into the structure and the dynamics of it, and not infer its properties from lack of signals, which can have many origins depending on the exchange rate, size of complex etc. Insight into the dynamics can be done either by titration of the complex, so that R1rho and/or R2s can be obtained under sub-saturation (5% bound, 10% bound, 20% bound) as done e.g. by the Blackledge group, or recording of CMPG.

More appropriately, it should be attempted to access the bound state structure by e.g. CEST to get the ¹³C chemical shifts under similar sub stoichiometric conditions. Running CD would also be a possibility to access if there are folding-upon binding of part of the N-Myc as suggested by the ITC. Is it not likely that the two transient helices of N-Myc would fold upon binding as they are involved in the contacts? A mutation to break the helices would add some information.

What are the protein concentrations in the HDX-MS experiments? It is stated that the complex is in a 1:3 molar ratio, but there is no information on concentration only volume. The ratio of 1:3 means that there are lots of free N-Myc present. Depending on the concentration, with a Kd of 1 micromolar, working e.g. at 1 micromolar concentrations will only produce about 40% bound state. Thus, the contribution to protection will be dominated by the free N-Myc.

If this is a disordered complex and if the complex is dynamic, effects on mutation of the different sites should be relatively independent. Thus, recording NMR relaxation on the variants that abolished some of the main sites (FYPD, FYFG and WKKF), would support this. It is interesting that in the context of the long N-Myc variants, there is less loss of intensity, suggesting a shift in the exchange regime by moving to higher affinity, again questioning the extreme dynamics of the complex.

Dynamic complexes with kinases have been shown before and is likely not as novel as suggested by the authors, and the dynamics in these kinase complexes serve different functional roles, e.g., facilitating multisite phosphorylations (e.g., Orand & Jensen 2025; Kragelj *Biomolecules* 2021; Hendus-Altenburger et al., *BMC biology* 2018) – is this the case here as well? Is N-Myc a substrate of active Aurora? Or does the first 100 residues of N-Myc serve as anchoring for substrate sites in the remaining N-Myc?

The presence of FXF sites in N-Myc resembles sites for ERK2 and JNK interactions as shown in Orand & Jensen 2025; Kragelj *Biomolecules* 2021; Hendus-Altenburger 2018 and more. Does N-Myc phosphorylation contribute to the activity assay conducted? Not clear how this assay works from the description in the manuscript. How does these sites in N-Myc compare to the FXFP sites for MAPK interactors generally observed, should be discussed (described by the Kornfeld and the Goose groups in the 90'ies). A discussion on motifs is missing.

Targeting kinases have shown quite dramatic side effects as these molecules have multiple partners. The suggestion of targeting the aromatic pockets to inhibit binding of N-Myc will also affect all other partners binding to this site. Thus, there are expectedly several side effects associated with such a drug scheme. Should be considered in the discussion (see e.g. work on JAK inhibitors)

The authors used the term “touchpoint” to describe the interaction. It is not clear what defines a touchpoint - is this an accepted term? They go on to mention “dynamic touchpoints”, how are they different from touchpoints? Why not use commonly used descriptions of interactions?

The abstract states this study goes against the “current dogma” – but the current dogma is not clear? Several disordered complexes involving IDPs have been described, and the current view is that retained disordered in complexes covers a continuum from fully folded upon binding to fully disordered. It requires some explanation and definitions to follow what the authors mean regarding “current dogma”.

Minors:

It is concluded that the stability of the complex correlate with affinity, but no correlation plots or statistics have been done to support this. Can this be included to support this conclusion? Is unfolding reversible?

The sequence in Fig. S3 should be included in the main figures, avoid spaces in the sequence, this is confusing.

Figure 5C, move data to SI and show the bar graphs with uncertainties instead.

Number of repetitions of the ITC should be at least $n=3$. Consider the numbers and the relevance (e.g. 3.98 ± 0.45 should be 4 ± 0.5). Uncertainties in the ITC should be on the figures

Why are both DDT and TCEP added?

More description of the kinase assay is needed to understand the experiment

Post transcriptional versus post translational – why use these two terms?

Line 173, some underline under Che should be removed

Line 183, what is residue motifs?

Line 477 $\alpha \rightarrow$

Line 478 (REF Diana)

Version 1:

Reviewer comments:

Reviewer #1

(Remarks to the Author)

The authors have thoroughly addressed all of my comments in this revision. The manuscript overall significantly advances our understanding of the interaction mode between AurA and N-Myc, and is suitable for publication in its current form.

Reviewer #3

(Remarks to the Author)

Hultman and coworkers have presented a revised manuscript where they have gone far to provide the additional experimental data requested which aide to support their conclusions. I appreciate the responsiveness of the authors to the suggestions made and in clarifying the text, not overinterpreting their findings. The manuscript is clearly improved with the new data, and with the attention to the textual formulation. However, I find that there are still a few issues and comments remaining that must be addressed:

There is some inconsistency in the SEC-SAXS and the ITC, that is not reflected upon. While SEC SAXS suggest a 1:1 complex, the ITC is probing a 2:1 complex (molar ratio 0.5). Weak binding as 5 micromolar are often difficult to capture in co-elution on the SEC, so what is the origin of this discrepancy? Other disordered complexes form higher order complexes, and this may be very relevant to address here as well. Some explanation is warranted. Can NMR data help here, in the sense that the peak movements – even at the border of the disappearing regions – may respond differently at the different stoichiometric ratios. This will add to the complexity of the binding mode, but given the disordered nature of it, perhaps not too surprising.

If you have multiple binding sites in N-Myc that are non-identical, binding to multiple non-identical sites on Aurora A (discussion p. 17, line 508), is it then allovalency? (see e.g. Locasale, Allovalency revisited: an analysis of multisite phosphorylation and substrate rebinding. *J Chem Phys* 128:115106. doi:10.1063/1.2841124). In my opinion, and according to the definitions, it is not. It appears instead to be a case of multivalency. Please reconsider and relate to the definitions of different binding mechanism described in the literature.

It seems that the region of MB0 needs adjusting based on the R2s and on the I/O – it appears to extend on the C-terminal side, please rethink this region. Are the R2 changes “dramatic” – or just what is expected from binding to a larger kinase? Please rethink.

Point-by-point response to Reviewers 1-3

We thank the Reviewers for their helpful, generous and critical advice, which we feel has significantly strengthened this manuscript. Importantly, the Reviewer comments have prompted the inclusion of new data that has further strengthened the conclusions of our work.

In summary, the new data include:

- A full set of fast relaxation experiments by NMR on N-Myc at several substoichiometric concentrations of Aurora A, where the very rapidly increasing R_2 supports a dynamic interaction. (**New Fig. 4**)
- NMR data of a 1:1 complex of N-Myc – Aurora A saturated to 93% still shows no recovery of signals line broadened by intermediate or fast exchange or increased molecular weight, which together with the rapidly increasing R_2 indicates that the interaction is dynamic on a time scale higher than the T2 timescale (0.1-0.2 s). (**New Supplementary Fig. 5**)
- A new set of HDX-MX experiments to analyse the complex at 1:1 stoichiometry and high saturation in order to evaluate the protection of bound N-Myc. We find that the protection observed is well below what would be expected for a stably bound complex, suggesting retained dynamics (**New Fig. 5E,F, Supplementary Fig. 5B**). This and previous data is jointly uploaded to PRIDE as described below and in the manuscript.
- As requested by Reviewer #3, we have increased the number of replicates for the ITC experiments and revised figures as advised; overall results were unchanged (**Fig. 6D and Supplementary Fig. 9**).

Further to this, we have included:

- a correlation plot, requested by reviewer #3, between the K_D 's and the ΔT_m s as presented in Fig. 5E showing a R value of 0.95 supporting the correlation between N-Myc interaction affinity and thermal stabilization of Aurora A (**New Fig. 5E**)
- Full SEC-MALS elution profiles as requested by reviewer #2 are now provided in **Supplementary Fig 1 and 3** supporting monodispersity of our investigated complexes.

The entire manuscript has been edited for clarity with special attention to items raised by the Reviewers. In response to Reviewers 2 and 3, we have carefully revised wordings referring to dynamic properties and decided to entirely avoid the term “fuzzy complex” even in the title due to some unsettlement still in the field regarding its definition in molecular terms. We are still convinced, and even more so with the new experimental data, that the complex is highly dynamic, especially as several orthogonal approaches strongly point in this direction. However, the collected argumentation and discussion on this is now entirely in the discussion.

Our point-by-point response to all Reviewer's comments is appended below.

With best regards, and for all co-authors,

Johanna Hultman Alexandra Ahlner Maria Sunnerhagen

REVIEWER COMMENTS

Reviewer #1 (Remarks to the Author):

In this manuscript, Hultman et al perform an integrative study of the N-Myc – Aur interaction, providing insights into the architecture of this complex and the molecular determinants for complexation. The manuscript is generally well constructed, and the set of experiments performed is logical. However, in places there is a lack of detail which may hamper a non-expert from being able to understand the data presented. My area of expertise lies in the application of HDX-MS and other MS tools for structural applications, and as a result my comments are primarily aligned with this portion of the work.

Reviewer #1, Comment 1:

- The authors perform extensive SEC-MALS and SEC-SAXS analysis. They present the resultant data in various ways – e.g. Guinier analysis (line 154) and $p(r)$ analysis (line 156). It would be helpful for the authors to provide further detail as to how the data presented support their conclusions. For example, this could be included in the appropriate figure legends. As written, for a non expert in these techniques it is hard to understand how the authors have drawn the conclusions presented.

Response:

We agree that the results from the SEC-MALS and SAXS data needed further explaining in order to guide the reader and make the conclusions more readily available.

We have now clarified the data interpretation in the main text, added an appropriate reference from the Svergun lab, and revised the figure legend of Fig. 2.

Reviewer #1, Comment 2:

- Why are the authors showing two runs of AurA data in Fig. 4C? This is confusing to the reader and is not discussed in the text/Methods. They are pretty well identical and im not sure both are needed to be shown here.

Response:

Due to the heavy use of the national infrastructure, the two HDX-MS experiments included in the first submission of this manuscript were independently scheduled and performed separately. To enable comparison between the experiments, corresponding free AurA states were run within each set of experiments. We appreciate that Reviewer #1 agrees with us that the free AurA runs are indeed highly similar, or even “pretty well identical”. Still, to ensure data integrity and correct interpretation, we would prefer to show the free AurA HDX-MS results for both experiments in Fig 5C (former Fig 4C) to emphasize the reproducibility, increase data transparency and allow the reader to make their own judgement based on the data provided.

To increase clarity for the reader, we have revised the main text, figure legend and annotations of Fig 5C (previously 4C). In the methods section, we have further clarified the experimental setup of the two experimentally equal but timely distinct HDX-MS runs.

Reviewer #1, Comment 3:

- More labels are needed in Figure 4. For example it is not clear to me the colouring shown in the top part of panel A. The authors state that the colouring here represents the innate HDX pattern, but that is a vague statement and further details are needed.

I assume this has to do with percentage uptake from maximal deuteration but the ‘key’ on the right of the figure should be labelled to indicate this.

Response:

Agreed. We have now clarified the figure legend to state that “The bars show the HDX-MS sequence coverage and are colored by the average deuterium uptake over all investigated timepoints”. The label “ % Deuterium uptake” has been added to the color key. To ensure clarity, we have followed this up with corresponding clarifications in Methods.

Reviewer #1, Comment 4:

The authors also need to state why they have chosen a '75 %' uptake (from Methods) to mean 100 % uptake for such an analysis. This is non-standard in the field from my experience.

Response:

In our setup, as in others, achieving 100% deuteration is impossible both due to the dilution of a ¹H₂O-based sample with D₂O (1:10) and as deuteration levels are commonly further reduced by back-exchange occurring within the system. To account for this, we have set the maximum deuteration level to 75% in HDEaminer and we ensure that no peptide exceeds 100% theoretical uptake. The set value of 25% for system wide back-exchange is based upon the experience of having used this particular HDX-MS approach for >200 HDX projects within the SciLifeLab Structural Proteomics Swedish national infrastructure over the last 5 years (for list of citations see <https://www.scilifelab.se/units/structural-proteomics/>). While this approach may result in a slight over- or underestimation of the actual uptake, it is important to note that this correction in HDEaminer only affects the color scale of the % deuteration heatmap. The comparative analysis is always based on relative comparisons of uncorrected deuterium uptake between bound and unbound states as determined from residual butterfly plots. Our approach allows us to obtain an accurate and reliable visualization of the overall protein without the need for doing a fully deuterated control. For the experts to scrutinize all details of our analysis, we have added all the visual exports from HDEaminer to the table in PRIDE, and in the manuscript we have significantly detailed the description of the Methods section for the HDX-MS experiments.

Reviewer #1, Comment 5:

Label scale bars shown in lower part of 4A and Figure 4B. Might also be useful to add labels to the structures shown in Figure 4D given that they refer to key regions in the structure in the text.

Response:

Well noted; we have revised Fig. 5 (previously Fig. 4) as proposed.

Reviewer #1, Comment 6:

- The authors state different residues are involved in interactions – e.g. 162-184 (line 239). But the data where protection is referred to doesn't directly correlate with these exact residue numbers (at least in terms of the labels shown). How were these numbers selected?

Response:

We thank the reviewer for pointing this out, and we agree that how the different residue stretches highlighted in our main text were selected was not made clear in the first manuscript version.

The protected regions were identified based on statistics as described in our response to Comment 7 below. Protection data is visually summarized in heatmaps, such as chiclet plots as shown in Fig. 5B, but we also show examples of uptake plots from key regions (Fig. 5C), and the total deuterium uptake over all investigated timepoints (Supplementary Fig. 6). Detailed examination of the HDX results, including the identity of each peptide in the analysis, can be

gained by the reader from the uptake plots for the individual peptides attached as supplementary files with the manuscript; these are also available on the PRIDE repository.

To help the reader traverse the difference between sequence labelling in the fragment map (Fig. 5A) and peptide # labelling in the figures where protection is visualised (Fig. 5B, Supplementary Fig. 6), we have revised the sequence labels in Fig. 5A to preferentially highlight the peptide segments in Fig. 5B that correspond to structural elements in AurA (Fig. 5A). In Fig. 5B, and related to Supplementary Fig. 6, we found a typo in the Fig. 5B label, where 208-225 should have been 208-236. This has now been corrected, and more thorough labelling has been added to the figures. In addition, we have revised the main text to better guide the reader and kept focus on our key findings.

Reviewer #1, Comment 7:

- How do the authors define 'strong' and 'weak' HDX protection (e.g. line 240 and Figure 4D)? This needs clarification and a metric applied. Was any statistical analysis used to identify protected regions?

Response:

To identify significant HDX protection, we used the built-in statistics in HDExaminer with the most stringent significant criterion. In the revision, this is now more carefully described in Methods, and the corresponding significance results for our study are explicitly described in detail in Table S1 available in the PRIDE repository.

To visualise the protected areas on the AurA structure in Fig. 5D (previously Fig. 4), we classified significant protection at early time points as 'strong' if it persisted over several of the time points measured, and 'weak' if it persisted over only one time point. Protected areas were only mapped onto the AurA structure if they were persistent in several peptide fragments in the same region, as shown in Fig. 5B. To distinguish between early and late protection, we choose a blue/pink color scheme. In the revision, we decided to not subclassify late protection as "strong" or "weak", as their patterns were quite similar. All this has now been clarified in the Main text and in the corresponding figure legend.

Reviewer #1, Comment 8:

- Given the difference in affinities, which the authors measure here, what is the difference in percentage bound during the HDX experiments for the two Myc constructs studied? This could be calculated and be helpful in interpreting the data showed. This should be stated in the text and a discussion of this included.

Response:

Based on the equilibrium K_D s obtained by ITC, the percentage bound in the two HDX-MS experiments after the 1:10 D_2O dilution was 87% (AurA Run1), 69% (AurA Run2). A higher saturation is obtained for the slightly stronger N-Myc binder. To be noted is that in the mix-and-dilute HDX experiment, a ten-fold dilution step into D_2O is an essential component of the HDX experiment, but as AurA alone is unstable at concentrations above 60 μM , together with the magnitude of the K_D s this limits the final concentration of the complex obtainable during exchange conditions.

In this revision, we included a third HDX-MS experiment with the aim to better characterise any protection on N-Myc in the complex. We here reached 77% bound in a 1:1 N-Myc₁₋₁₀₀: AurA complex. All the information above is included and briefly discussed in text and methods.

Reviewer #1, Comment 9:

- The authors display an output from PEAKS in Supplementary Fig. 4 where putative phosphorylation sites are indicated. In my experience, PEAKS can annotate very low confidence PTM sites in these types of reports, where the MS/MS data do not provide the requisite support for the presence of PTMs at a given residue. The authors should investigate their data further, and include MS/MS spectra for all phosphosites that are confidently identified. I would recommend changing this figure to one that only shows the localisation of confident phosphosites after this manual data curation has been performed.

Response:

As the reviewer correctly states PEAKS can be a bit optimistic in assigning PTM's, however, this was not the main purpose of this analysis. Rather, it was discovered that there was a lack of HDX peptide coverage in the AurA active loop region, which is known to hold phosphorylation sites at T287 and T288 (Bayliss et al., Mol Cell 2003). The phosphorylation analysis was done on the undeuterated MS/MS samples used for generation of a HDX peptide pool and helped identify peptides in this region. For the purposes of increasing HDX-MS sequence coverage, we therefore show the "uncorrected" coverage as it is not needed to have 100% secure site assignment. Furthermore, as stated in the text there was no significant difference between the protection of phospho- and non-phospho peptides. To address any possible effects on N-Myc binding from AurA active loop phosphorylation, we designed AurA mutants and found that phosphorylation at these sites has very little or no effect on N-Myc binding (Supplementary Fig. 7B). A dedicated phosphosite analysis would need to have been done using ETD on a different MS instrument, but given our results at this level of analysis and the already well-established AurA phosphorylation pattern for both eucaryotic and *E. coli*-produced protein, we found such analysis to be of little relevance to the conclusions in this work and therefore outside the scope of this study.

We have revisited the data, curated it manually and based on this analysis revised Supplementary Fig 7. In this revision, we have revisited the data, curated it manually and based on this analysis revised Supplementary Fig. 7A to include only the phosphorylation sites in the region spanning the AurA active loop. In the text, we have clarified that this analysis was done merely for the purpose of increasing peptide coverage and not to determine the exact phosphorylated residues. The data is included in deposited files in PRIDE.

Reviewer #1, Comment 10:

- In the methods indicate the protein concentrations used – this is critical for understanding the percentage bound in these experiments.

Response:

We agree with the reviewer; this was an unfortunate omission on our side. We have now added the protein concentrations to the text including an estimated stoichiometry based on the affinities measured by independent techniques (ITC and BLI). See also our response to Comment 8 above.

Reviewer #1, Comment 11:

- What do the authors mean by the FASTA file only containing the RDB sequence (line 703)?

Response:

This is an error on our side; the text should read "the targeted protein sequence" instead of "the RGB sequence". This has now been corrected.

Reviewer #1, Comment 12:

- The authors must indicate the PRIDE dataset identifier in their manuscript – at the moment this is missing (line 699)

Response:

The PRIDE dataset was deposited on the Editor's request after our recommended manuscript transfer from Nature SMB to Nature Comm was completed. At that stage we could not make any alteration to the already submitted manuscript text, but we submitted the update with PRIDE identifiers within the Data Code Availability pdf as required, and assumed this would also reach the Reviewers. The data set has been in the PRIDE database since March 4th. In the current revision, we have now been able to properly include this information in the Data Availability Section. We have also deposited the new HDX-MS data recorded during the revision process under the same identifier.

Reviewer #1, Comment 13:

The authors must include experimental details for their phosphosite analysis.

Response:

This has now been described in detailed in Methods under the HDX-MS section and is mentioned in Results, please see our response to Comment 9 above.

Reviewer #1, Minor comments:

Line 121 – add 'mass' before spectrometry.

In several places references are missing. E.g. line 478 and line 715. Please thoroughly proof-read the manuscript.

- We apologize for these oversights and thank the reviewer for pointing this out. We have now corrected these errors and thoroughly proof-read the revised manuscript.

Reviewer #2 (Remarks to the Author):

The N-Myc MB0-MBI region forms a specific, functional and fuzzy complex with the N-lobe of Aurora kinase A, Hultman et al.

The manuscript describes the interaction between N-Myc and Aurora A. Disordered protein regions, like those found in Myc, are increasingly being viewed as potential drug targets. Given its near ubiquitous role in many cancers, Myc is frequently cited as a potential target. Finding interfaces that are formed when an IDR binds to another protein are an attractive alternative to simply trying to identify small molecules that can bind to IDRs in isolation. The work presented here not only does a great job in characterizing the specific interaction between Myc and Aurora A, but also lays out a straight-forward (albeit using a number of biophysical techniques) approach for characterizing such interfaces in general. The manuscript is written in a clear and succinct fashion and is easy to read and follow. The figures are all well organized and clear (I particularly appreciated the unified color things throughout the entire paper).

The authors use a combination of multiple, complementary, biophysical techniques that both report on structural features and affinities. Given what is required to obtain good quality NMR spectra on a protein like Aurora, I appreciated the use of the HDX as a more cost-effective and simpler (albeit lower resolution) alternative. With some revisions, this manuscript is certainly worthy of publication.

Reviewer #2, Comment 1:

The use of “fuzzy” and “dynamic” to describe the complex, while quite possibly true, is not heavily supported in the data presented. These terms explicitly describe a dynamic process, and the vast majority of data in the paper is ensemble averaged and does not directly report on dynamics. The one exception is the HDX. I think the interpretation of the time scales of protection do hint at some dynamics in the complex, but the long time point (9000 s) is really quite long and I’m not sure how these exchange dynamics would compare to any complex, dynamic or no, with micromolar affinity.

Response:

We thank Reviewer #2 for appreciating our work, including our choice of HDX as an option to analyse binding effects on AurA. As explained, we used HDX to identify any protection patterns on/in AurA on binding MYC, and, as the reviewer mentioned, we also recognize that HDX will not resolve local flexibilities on ps-ms timescales. However, HDX has been well validated and appreciated as a technique able to capture changes in dynamics in a time window of seconds that pertains to both folding and collective motions within, or between, larger protein domains (Henzler-Wildman & Kern, Nature 2007, Masson et al., Nature Methods 2019 and more). Having said this, we do agree that the longest time point (9000 s) is “really quite long” and in our interpretation of data we therefore do not consider any protection/deprotection that only relies on this, or any single, time point. In response to Reviewer #2, we have clarified in this manuscript re-submission that our definition of strong and weak effects in HDX requires significant differences in deuterium uptake over at least two time points.

Reviewer #2, Comment 2:

The authors also cite other NMR studies of IDPs (such as Milles et al) to suggest that their NMR data is in agreement with other studies of dynamic complexes. What is missing in this comparison is the careful NMR relaxation dynamics experiments that are present in those papers. If the authors were to include R1 R2 hetNOE and relaxation dispersion analysis, the point would be significantly stronger. This data would also significantly enhance the paper's conclusions.

The authors conclude that the significant line broadening on addition of Aurora is the result of intermediate exchange (and thus also indicative of a dynamic complex). While it is possible that this is true, a more likely explanation is simply that residues in complex with Aurora are tumbling with a much different τ_c and are broadened relative to the other disordered residues. This is quite common in complexes of IDRs with larger, folded, binding partners regardless of the exchange regime. In order to make this point, again, a much more detailed accounting of the NMR relaxation should be done.

Response:

We agree entirely with the Reviewer that a comprehensive NMR dynamic evaluation would increase our understanding of the highly clinically relevant N-Myc-AurA complex. However, with N-Myc-AurA, our accessible experimental conditions are quite limited. Due to the very significant line broadening on AurA binding, N-Myc resonance signals in binding regions as defined by CSPs are lost already at 10% AurA and do not recover at saturation, which precludes direct NMR analysis of N-Myc dynamics in the bound state. In the relaxation analysis that is done in for example Milles et al (2018) significant signal remains at 20% saturation, enabling their NMR studies. To explicitly demonstrate the effects of N-Myc line broadening to the reader, the NMR data is now complemented by a ^{15}N - ^1H HSQC for the 1:1 complex of ^{15}N N-Myc₁₋₆₉ and unlabeled AurA, showing that N-Myc peaks in the binding regions are not recovered at near-

saturation (93%, 50 μ M of each component) well above the K_D (4 μ M) of the complex, performed at the highest concentration obtainable for AurA in buffers and temperature where N-Myc is also happy.

Within this experimentally accessible window, we were still able to determine NMR fast relaxation rates for all N-Myc residues with substoichiometric amounts of AurA, as suggested by both Reviewers 2 and 3, but only up to a degree of saturation of 6% AurA, after which some signals were too small to evaluate reliably. These data, recorded at 600 MHz, are now included in the Revision. We find, and show, that while N-Myc R_1 s are not much affected by the addition of AurA in agreement with R_1 rates in general being small for 15 N in biomolecular systems, R_2 s are very much increased as expected for a system in intermediate to fast exchange on the T_2 timescale. (Baldwin and Kay, Journal of Biomolecular NMR, 2013)

As the reduction of the signal is much higher than proportional to the amount of ligand added, we can exclude that the exchange is slow. If the system is in intermediate or fast exchange with respect to T_2 relaxation, resonances will be broadened beyond detection because of exchange broadening owing to the much smaller and dominant T_2 of the larger component, the theoretical background of which has been carefully described elsewhere by the Perham group (Howard et al, JMB 2000). With respect to the R_2 s, we therefore find ourselves caught in a dynamic region where we, at this stage, will not be able to evaluate T_2 s to the level required for a complete dynamic analysis, and we are also not able to measure R_1 s or hetNOEs for the 1:1 complex due to the extreme line broadening.

What we can conclude is that N-Myc is in intermediate or fast exchange with respect to T_2 relaxation, but we cannot define whether such exchange is only between the free and one bound state or if this also includes exchange between several bound states (avidity). However, deletion of the third C-terminal N-Myc interaction with AurA (as in N-Myc₁₋₆₉) has very similar effects on AurA as judged by HDX, and furthermore, this three-touchpoint rapidly exchanging system migrates as a 1:1 complex on a gel filtration column. This suggests that the three N-Myc interacting regions bind AurA and can at least partly compensate for each other with regards to AurA binding, suggesting allovalency and/or avidity. This is also independently concluded from the SAXS-based modelling of the complex, although at low resolution. While we are committed to resolving the dynamic features of this complex in detail, this is outside the scope of the current manuscript as it will require comprehensive NMR experimental optimization (temperatures, magnetic fields and pulse programs) and theoretical analysis, together with significantly extended mutational analysis and careful biophysics.

With all this in mind, and while taking care not to overinterpret data, in this revision we have included, and commented on, the fast relaxation results in the Main. In the discussion, we comment on the plausible allovalency within the N-Myc - AurA complex and its agreement with our other biophysical data, as a basis for further investigation.

Reviewer #2, Comment 3:

The conclusion that the complex is 1:1 is based on SEC-MAL measurements where the starting stoichiometry was only 2:1. It isn't clear to me that either 1. There isn't evidence of a higher MW complex at the front edge of the complex elution peak or 2. That there couldn't be a higher molecular weight complex where a second MYC (or second kinase) is added with a much lower affinity. For example, this paper from Ben Schuler's group describes an archetypal dynamic complex:

<https://www.nature.com/articles/nature25762> .

The original work concluded it was a 1:1 complex. However, later investigation at different concentrations revealed the existence of trimers and even phase separated droplets: <https://www.nature.com/articles/s41586-023-06329-5> . I think that it is likely fair to conclude that a 1:1 complex largely dominates the measurements shown here, but I don't know if its reasonable to conclude that a 1:1 complex is the only complex that can form. I agree, with the authors that the SAXS data as shown also suggests a 1:1 complex. However, we do not see what the elution profiles look like for these data and the same concentration dependence would apply here as does for the SEC-MALS.

Response:

SEC-MALS elution profiles are now included in Supplementary Fig. 1 and 3, and SEC-SAXS elution profiles in Supplementary Fig. 2 and 3 with the respective SAXS-scattering data (alsodeposited in SASBDB). None of these profiles show any indication of a heavier complex with similar or higher affinity. The SAXS experiment is very sensitive to formation of higher-molecular weight assemblies but shows no such indications in the scattering curve (linear at low q) or Guinier analysis (molecular weight corresponds to 1:1), or in the $P(r)$, which goes nicely down to 0 at longer D_{\max} .

As in every biophysical investigation, we cannot exclude that much weaker additional binding events occur. However, any biological effects of such binding would require unphysiologically high concentrations of both N-Myc and AurA. As described above, detailing significantly lower affinities than the low-micromolar K_D s identified there would be experimentally challenging due to aggregation at higher concentrations as described above. For completeness, being in the IDP community, we are of course aware of the possibility that phase separated droplets could occur when working with MYC, but we have not seen any indications of such in the N-Myc – AurA system.

The limitation of AurA concentration has been included in Methods and is referred to in the main text.

Reviewer #2, Comment 4:

The nano DSF data is very compelling. It is very clear that it correlates with the HDX and show specific residues that are critical to the interactions. It is also great to see that this correlates with activity. One minor comment is that this should be referred to as “DSF” data and not “Prometheus” data in the text.

Response:

We acknowledge the Reviewer's appreciation of the nanoDSF data, and we agree with the reviewer that we should not refer to the technique by the brand name. We have changed the text to nanoDSF and refer to a methods review of the technique to assist the reader.

Reviewer #2, Comment 5:

I liked the structural modeling section. Perhaps I have missed this in the literature, but, to my knowledge, this is the first use of AF structures in this way. That said, I think the more appropriate way to have approached this (as opposed to sorting by how well individual structures fit the experimental data) would be to use one of the methods that is established in the IDP field for fitting saxs data to ensembles. Either the EOM method included in the ATSAS package or any type of ensemble reweighting (<https://pubs.acs.org/doi/10.1021/acs.jctc.8b01231> for example). Given that the structural details seemed quite similar for the full ensembles versus the filtered, I don't suspect there

would be a huge difference in the conclusions for this paper, but the method would be more appropriate and the final results could be a bit cleaner.

Response:

We thank the reviewer for raising this important point. We are, of course, aware of established ensemble approaches such as EOM and ensemble reweighting (see our work in Caporaletti et al., *Biophys J* 2023; Salomonsson et al., *Protein Science* 2024), and we also tested such methods in our analysis. In fact, when applying reweighting across the full pool of ~10,000 AlphaFold-generated models, we could reach χ^2 values essentially at 1.0. However, in this case, we judged this approach to be heavily overdetermined, since the SAXS data provides only limited constraints and a sufficiently large pool of conformers allows almost any profile to be matched extremely well. For this reason, we deliberately chose a stricter criterion where only those individual conformations with $\chi^2 < 1.1$ were retained. This procedure effectively defines an ensemble that is consistent with the SAXS data, while avoiding potential artifacts of excessive reweighting. We view this strategy as more conservative and transparent, since it provides a direct link between physically plausible models and experimental agreement, instead of relying on an optimized weighted mixture over thousands of possibilities. In our revision, we have pointed this item out in Methods, and, briefly, in Main.

Reviewer #2, Comment 6:

Finally, I think that it is worth noting that the introduction contained a great level of detail that keys the reader in to why this particular interaction could be important in a broader biological setting. This is not always the case in biophysics papers.

Response:

We thank the reviewer for recognizing our effort on putting our work in a broader biological context. In the revision, due to lack of space, we have however had to slightly reduce some detail in the introduction, but we hope this is still sufficient to key the reader in the right direction as the Reviewer suggests.

Reviewer #2, A few small details

“most human cancers” in the introduction – I would prefer most specifics

- MYC is indeed overwhelmingly critical for nearly all human cancers, as recently reviewed in *Nature Rev Clin Oncol*, and at least 27 out of the 33 human cancer types in the Cancer Genome Atlas are associated with MYC (new references added in revision). As this is a molecular study, there is limited space to expand on this with more clinical perspectives. To meet the Reviewer, we have polished this statement and added two comprehensive review articles to support it, but for specifics we will have to refer the reader to selected reviews.

Line 121 “several additional techniques” – there aren’t many more to list, perhaps just list them

- Completed

145 – MYC is an IDR based on DLS? Also, based on the long retention time on the SEC column, presumably the SAXS profile and the NMR HSQC peak dispersion

- Indeed; yes we have included these as well in the same context.

163 – SAXS Kratky analysis is referenced, but is not explained

- A short explanation has been included in the Supplementary/Methods

167 – It is worth noting that the two MYC constructs overlay nearly perfectly, also supporting the fact that they are disordered and there are no significant long-range interactions

- Indeed, we have included this in the revision.

478 Missing ref?

- Corrected.

Reviewer #3 (Remarks to the Author):

Reviewer #3; Comment 1:

The manuscript by Hultman and coworkers show an integrative structural biology approach to describe the complex between the first 100 residues in the transcription factor N-Myc and the Aurora A kinase. They conclude that the complex is “fuzzy” at 1:1 stoichiometry and formed with a strength of 1-4 micromolar. They show that binding to Aurora A leads to its activation and that binding can be inhibited by a competitive binding partner. The data have been carefully collected, and the manuscript is well written and provide new insight into the system under investigation. However, I have some reservations about the conclusion they reach - that the data supports a complex of multiple exchanging bound states; a conclusion – if true, that will need much more data to be confirmed, as detailed below. Also, there are already some studies on dynamic IDP – kinase interactions that have not been mentioned and referred to, but which are highly relevant for this study.

Response:

We thank the Reviewer for appreciating our integrated structural biology approach. N-Myc binding to AurA is a critical system for cancer research and holds high therapeutic potential but has hitherto resisted biophysical investigation in solution. We are committed to explore and understand its molecular mechanisms.

Reviewer #3; Comment 2:

My main reservation is related to the statement of this complex being dynamic and fuzzy. This conclusion is reached from disappearance of the peaks from the NMR spectrum and lack of protection against exchange monitored by HDX-MS. This does not necessarily mean that the complex is disordered. The problem is that we cannot observe the bound state (see comments to HDX-MS below). First, there are no experimental data on the dynamics (with is provided by rates) of this complex, so how can the authors interpreted it as dynamic? Not forming secondary structure in binding can be a case where no protection will emerge in HDX MS.

Response:

We appreciate the Reviewer’s request for clarity in this respect. The reviewer is correct in that we had weak HDX-MS information about the dynamics of the bound state due to the presence of free N-Myc in the HDX-MS runs in the submitted manuscript (1:3 ratio AurA:N-Myc). In the presented experiment it is true that the excess of free N-Myc would make it appear as largely unprotected in HDX-MS. However, given the large and clear perturbation on AurA it was still very surprising that N-Myc appeared to be so dynamic while 1/3 is bound. On the contrary, if N-Myc would have been a strong binder with a low koff it should be possible to observe a bimodal (1:3) isotope distributions for the interacting peptides, but this was not seen for N-Myc. With experience from hundreds of projects over 8 years, our HDX-MS expert Dr Simon Ekström at the Swedish National infrastructure (<https://www.scilifelab.se/units/structural-proteomics/>) has done quite a few IDP interaction studies and bimodality is something he has seen in many HDX-MS IDP interaction projects before, and where there is a stronger interaction, it has even been possible to deduce binding stoichiometry. The HDX data presented here are actually the first

time ever that Dr Ekström has seen an IDP that retains this high uptake while bound; indeed, even peptide binders usually have a clearer protection than observed for here for N-Myc.

In response to the reviewer, we have now complemented the HDX-MS data with a run at a 1:1 ratio N-Myc:AurA at 77 % saturation. While the same pattern of large HDX-MS perturbations were observed on AurA, only very weak perturbations were observed on N-Myc and only at the shortest time point. Essentially, we see the same thing for N-Myc as in our first set of experiments: regardless of free or bound form, the uptake is very high over the entire sequence. This is a clear sign that from an HDX perspective, N-Myc has a very high off rate while maintaining a high “occupancy” presumably due to high association rate on AurA, which agrees with the large observed perturbation of Δ HDX for AurA (1-3 Da) while Δ HDX for N-Myc is significantly lower (0.2-0.6 Da).

Of note, the effects on AurA as judged from HDX are very similar for N-Myc₁₋₆₉ and N-Myc₁₋₁₀₀. This implies that including the third binding site in N-Myc as evident by NMR I/I0 and CSPs does not seem to add another binding site on AurA, suggesting possible allovalent properties in this interaction. In agreement, the SAXS-based ensemble model suggest that all N-Myc binding regions contact the same regions on AurA irrespective of # binding sites, and the SAXS data is also not consistent with any single conformation where all binding regions on N-Myc are simultaneously bound to AurA. Together with the observation that the N-Myc-AurA complexes studied here elute as 1:1 on a gel filtration column despite the low-micromolar affinities observed, this suggests allovalency which is a heterogeneous binding model common to dynamic IDP interactions. Together with the orthogonal evidence from other methods clearly pointing in the same direction, this is a clear hallmark of a dynamic interaction.

Responding to Reviewer #2, in this revision we have included a new HDX-MS data set further validating the statement of a dynamic complex as described above and present the conclusions in Results. To help the reader, in the Discussion we now review all the data in our study that support allovalent binding properties of this interaction, encouraging the Reader to make their own judgement.

Reviewer #3 Comment 3:

The disappearance of the NMR peaks may simply be due to exchange broadening between two states – a free and a (single) bound state – caused by the exchange rate being in the intermediate exchange regime for NMR and not due to the presence of multiple states in the bound state (fuzzy). The conclusion that these regions “shows extensive dynamics” are not supported by the data.

It will be necessary to record data on the bound state to provide insight into the structure and the dynamics of it, and not infer its properties from lack of signals, which can have many origins depending on the exchange rate, size of complex etc. Insight into the dynamics can be done either by titration of the complex, so that R1rho and/or R2s can be obtained under sub-saturation (5% bound, 10% bound, 20% bound) as done e.g. by the Blackledge group, or recording of CPMG.

More appropriately, it should be attempted to access the bound state structure by e.g. CEST to get the ¹³C chemical shifts under similar sub stoichiometric conditions.

Response:

We agree entirely with the Reviewer that a comprehensive NMR dynamic evaluation would increase our understanding of the highly clinically relevant N-Myc-AurA complex. However, with N-Myc-AurA, our accessible experimental conditions are quite limited.

Due to the very significant line broadening on AurA binding, N-Myc resonance signals in binding regions as defined by CSPs are lost already at 10% AurA and do not recover at saturation, which precludes direct NMR analysis of N-Myc dynamics in the bound state. In the relaxation analysis that is done in for example Milles et al (2018) or Delaforge et al (2018) significant signal remains at 20% and 10% saturation, enabling their NMR studies. To explicitly demonstrate the effects of N-Myc line broadening to the reader, the NMR data is now complemented by a ^{15}N - ^1H HSQC for the 1:1 complex of ^{15}N N-Myc_{C1-69} and unlabeled AurA, showing that N-Myc peaks in the binding regions are not recovered at near-saturation (93%, 50 μM of each component) well above the KD (4 μM) of the complex. These experiments were performed at the highest concentration obtainable for AurA in buffers and temperature where N-Myc is also happy.

Within our experimentally accessible window, we were still able to determine NMR fast relaxation rates for all N-Myc residues with substoichiometric amounts of AurA, as suggested by both Reviewers 2 and 3, but only up to a degree of saturation of 6-10% AurA, after which some signals were too small to evaluate reliably. These data, recorded at 600 MHz, are now included in the Revision. We find, and show, that while N-Myc R_1 s are not much affected by the addition of AurA in agreement with R_1 rates in general being small for ^{15}N in biomolecular systems, R_2 s are very much increased as expected for a system in intermediate to fast exchange on the T_2 timescale. (Baldwin and Kay, Journal of Biomolecular NMR, 2013)

As the reduction of the signal is much higher than proportional to the amount of ligand added, we can exclude that the exchange is slow. If the system is in intermediate or fast exchange with respect to T_2 relaxation, resonances will be broadened beyond detection because of exchange broadening owing to the much smaller and dominant T_2 of the larger component, the theoretical background of which has been carefully described elsewhere by the Perham group (Howard et al, JMB 2000). With respect to the R_2 s, we therefore find ourselves caught in a dynamic region where we, at this stage, will not be able to evaluate T_2 s to the level required for a complete dynamic analysis, and we are also not able to measure R_1 s or hetNOEs for the 1:1 complex due to the extreme line broadening.

What we can conclude is that N-Myc is in intermediate or fast exchange with respect to T_2 relaxation, but we cannot define whether such exchange is only between the free and one bound state or if this also includes exchange to several bound states (avidity). However, as described in "Response to R3;2" above, the presence of two (N-Myc_{C1-69}) or three (N-Myc_{C1-100}) interacting regions has very similar effects on AurA as judged by HDX-MS, and furthermore, both these constructs similarly migrate as 1:1 complexes on a gel filtration column as detected by SEC-SAXS. This suggests that the N-Myc interacting regions can at least partly compensate for each other with regards to AurA binding, suggesting allovalency and/or avidity. While we are committed to resolving the dynamic features of this complex in detail as proposed by Reviewer #2, this is outside the scope of the current manuscript as it will require comprehensive NMR experimental optimization (temperatures, magnetic fields and pulse programs) and theoretical analysis, together with significantly extended mutational analysis and careful biophysics.

As advised by Reviewers #2 and #3, in the revision we have now added new NMR data showing dramatically increased N-Myc R_2 s under sub-stoichiometric AurA conditions (2, 6 and 10%), suggesting intermediate to fast exchange relative to the T_2 relaxation timescale.

In the discussion, we comment on the plausible allovalency within the N-Myc - AurA complex and its agreement with NMR our other biophysical data, as a basis for further investigation. Taking care not to overinterpret data, and in the absence of a full CPMG/CEST NMR analysis, we have toned down the conclusions regarding the dynamic state of the interaction throughout the manuscript.

Reviewer #3 Comment 4:

Running CD would also be a possibility to access if there are folding-upon binding of part of the N-Myc as suggested by the ITC. Is it not likely that the two transient helices of N-Myc would fold upon binding as they are involved in the contacts? A mutation to break the helices would add some information.

Response:

Indeed, as judged by CheSPI there is a very small propensity for helicity in segments comprising residues 48-53 in N-Myc₁₋₆₉, as well as in residues 75-86 in N-Myc₁₋₁₀₀. In the longer N-Myc fragment, the helicity in residues 48-53 in N-Myc₁₋₆₉ is slightly decreased and no longer validated by the CheSPI DSSP prediction (Supplementary Fig. 4). In the complex, as judged by NMR (very small CSPs) and HDX-MS (no reduced accessibility in N-Myc), the two possibly transient helical segments do not seem prone to folding-on-binding. By CD, the already very large amount of helix in AurA would make it very difficult to measure a small helical increase on binding, and even if such changes would be measurable, they would not be possible to directly attribute to N-Myc as CD cannot distinguish between helicity increase in the two binding partners. Furthermore, as the MBI region is extremely conserved across MYC homologues, any designed mutation here, and in particular to proline, would likely affect N-Myc interactions in unforeseen ways, rendering such an experiment inconclusive to address the Reviewer's concerns.

To respond to the Reviewer's concerns regarding whether there is folding-upon-binding of part of the N-Myc, we have for this revision performed HDX-MS experiments optimized for detecting N-Myc in the bound state. This method has been validated to recognize the formation of helical structure in INCENP binding to Aurora B, where significant changes in the HDX-MS pattern of the INCENP "IN-box" peptide revealed folding-on-binding onto the N-lobe (Segura-Peña et al., eLife 2023). In contrast, our N-Myc optimized HDX-MS measurements on N-Myc – AurA performed at 77% saturation in a 1:1 molar ratio shows very similar rapid H/D exchange in the free and bound states of N-Myc, which indicates that N-Myc folding-on-binding into secondary structure is not a primary mechanism here (see also "Response to comment 2" above). While this may seem contradictory to our ITC measurements, we need to keep in mind that ITC cannot resolve between these effects on a molecular level, as calorimetry considers the system as a whole.

In the revised text, the comparison with INCENP mentioned above has been introduced into the manuscript, and the text regarding transient helical segments has been revised to be more precise. To avoid overinterpretation, the thermodynamic profiles are now only pointed out as being similar, leaving out any molecular interpretations as these can not be distinguished from the thermodynamics alone

Reviewer #3 Comment 5:

What are the protein concentrations in the HDX-MS experiments? It is stated that the complex is in a 1:3 molar ratio, but there is no information on concentration only volume. The ratio of 1:3 means that there are lots of free N-Myc present. Depending on the concentration, with a KD of 1 micromolar, working e.g. at 1 micromolar concentrations will only produce about 40% bound state. Thus, the contribution to protection will be dominated by the free N-Myc.

Response:

We have now provided details on the saturation levels in the HDX-MS experiments, including concentrations and level of saturation, both in Methods and in Results. The HDX-MS experiments optimized to detect AurA protection, as described in the first submission, were performed at three-fold excess of N-Myc at concentrations ensuring saturation levels on AurA at 69 and 87% for N-Myc₁₋₆₉ and N-Myc₁₋₁₀₀, respectively, in the D₂O-diluted sample intrinsic to the HDX experiment. Thus, the contribution to AurA protection will be dominated by bound N-Myc, despite the experimental limitations imposed and the solubility limit for AurA (50 μM).

Furthermore, in this revision, and to resolve any protection of N-Myc contributed by AurA binding, we have included an additional HDX-MS experiment at 1:1 ratio of N-Myc to AurA, at a concentration ensuring 77% saturation. The protection of AurA is very similar, and the experiment allowed us to evaluate the possible protection of N-Myc in the complex with very little contribution from free N-Myc.

Reviewer #3 Comment 6:

If this is a disordered complex and if the complex is dynamic, effects on mutation of the different sites should be relatively independent. Thus, recording NMR relaxation on the variants that abolished some of the main sites (FYFD, FYFG and WKKF), would support this.

Response:

We are not sure what the Reviewer means by that the effects should be “relatively independent”. We also do not state or believe that the complex is entirely disordered; we apologize if our writing has been unclear here. However, we do agree that IF the complex would have been entirely disordered, it would – according to current knowledge in the field – most likely be driven by electrostatics and allovalency, and if so mutating single interaction sites would not affect binding. But in our case here, such mutations reduce affinity significantly and impairs AurA activity. The reduced affinity would make conclusive NMR relaxation on the variants studied here even more difficult. While further mutational and dynamic analysis of this interaction is a worthy objective, given the hardships of studying this particular system by NMR, we believe that such studies need to be very carefully designed to answer specific questions in a separate study.

Reviewer #3 Comment 7:

It is interesting that in the context of the long N-Myc variants, there is less loss of intensity, suggesting a shift in the exchange regime by moving to higher affinity, again questioning the extreme dynamics of the complex.

Response:

Here, we respectfully disagree with the Reviewer as it is not clear to us which residues Reviewer#3 is looking at. Within MB0 and MBI, the reduction pattern in I/I_0 is very similar in the two constructs, with final intensities differing by <5% in the few cases where there is still any intensity remaining at 0.3 mol equivalents of AurA. Outside MB0 and MBI, in contrast to Reviewer statement there is a notable (10-25%) higher loss of intensity in the longer construct. We propose that this increased loss of intensity in N-Myc₁₋₁₀₀ could result from higher affinity (K_D 0.9 μM vs 4.0 μM) together with increased tumbling time due to increased M_w and significantly increased apparent volume in the N-Myc₁₋₁₀₀ complex as shown by SAXS (R_g and Porod volume; Suppl Table 1) and SEC-MALS (increased elution volume; Fig. 3A).

Reviewer #3 Comment 8:

Dynamic complexes with kinases have been shown before and is likely not as novel as suggested by the authors, and the dynamics in these kinase complexes serve different functional roles, e.g., facilitating multisite phosphorylations (e.g., Orand & Jensen 2025; Kragelj Biomolecules 2021; Hendus-Altenburger et al., BMC biology 2018) – is this the case here as well?

Response:

We apologize for our unclear writing in the Discussion. It was certainly not our intention to overlook previous critical dynamics work in the kinase field; indeed, we bring up several examples of such dynamic IDP investigations later in the paragraph, including the suggested Orand & Jensen (2025) as well as Csizmek (2017) and, earlier, Xie et al (2020). In this revision, we have also included the comprehensive review by Camacho-Zarco et al (Chemical Reviews, 2021) which focuses on NMR evaluations with several examples of kinases.

However, our aim in this paragraph of the discussion was to review cases where IDP interactions have been shown to affect enzyme activity, as to date a majority of the IDP interactions characterized at molecular and structural level seem to effect biology by serving as a scaffold to bring proteins together to form a complex, rather than by directly activating enzymatic function. Indeed, thanks to works in the groups of Blackwell and Jensen pointed out by Reviewer#3, the molecular interaction landscape of mitogen-activated protein kinases (MAPK) kinases has been resolved in structural and dynamic detail. However, Aurora kinases belong to the distinct protein kinase A (PKA) family, as structurally resolved in groundbreaking works by the Taylor group and others. Within this subgroup, the Aurora kinases have quite specific structural and dynamic features, including a distinct regulatory spine as described by the Levinson group. Furthermore, in contrast to MAPKs, Aurora kinases have been shown to be predominantly activated through interactions with its N-terminal lobe as well as through ROS-enabled cysteine-governed dimerization. Direct comparisons with the MAPKs and their interaction motifs are therefore difficult to do and would not add to the understanding of the N-Myc-AurA system in the current context, although we agree that this would be well considered in a future review on kinase-IDP interactions.

Due to space constraints given the added data required by reviewers in this revised version of the manuscript, we have been forced to limit our referencing to include only the most relevant kinase complexes both in the introduction and discussion, and primarily need to focus on referencing reviews. However, related to this we agree with Reviewer #3 that it is relevant to discuss the function of dynamics in complexes with Aurora A including possible binding motifs; for this please see our response to Reviewer #3 Comment 10. In the discussion, we also dedicate a MYC-centred paragraph to the discussion of tunable interactions, such as by phosphorylation, and how this could play significant functional roles in shifting the populations of alternative binding modalities.

Reviewer #3 Comment 9:

Is N-Myc a substrate of active Aurora? Or does the first 100 residues of N-Myc serve as anchoring for substrate sites in the remaining N-Myc?

Does N-Myc phosphorylation contribute to the activity assay conducted? Not clear how this assay works from the description in the manuscript.

Response:

N-Myc is not a known substrate for AurA, within the first 100 residues or elsewhere. The consensus phosphorylation motifs for AurA are [R/K/N]-R-X-[S/T]-B, where X stands for any

amino acid and B stands for any hydrophobic residue [AFILMV] except Pro, reported by Ferrari et al (Biochem J 2005)¹, and R-X-S/T-[ILV] reported by Ohashi et al (Oncogene 2006), neither of which is present in the N-Myc sequences used here. However, c-Myc has recently been identified as a substrate for Aurora B, with phosphorylation at c-Myc-Ser67 (Jiang et al., Cancer Cell 2020). This c-Myc serine, boldfaced below and directly C-terminal to MBI, is not conserved in N-Myc:

c-Myc : 58-TPPLSPSRR**S**GLC

N-Myc : 58-TPPLSPSRGFAEH

The AurA kinase assay was performed with the ADP-Glo™ kinase Assay from Promega Corporation (Madison, USA) to determine the activity of our recombinantly produced AurA. During the first step of the activation assay, the kinase is incubated with substrate, ATP in a buffer including MgCl₂, in the presence or absence of N-Myc, and the remaining ATP after completion of this reaction is depleted. The generated ADP from this step is then converted to ATP and quantified using the luciferase/luciferin reaction. This has all been detailed in Methods.

Of note, none of our biophysics experiments are conducted in the presence of ATP, thus, phosphorylation will not be possible during our NMR, SAXS, HDX-MS or biophysics experiments.

Reviewer #3 Comment 10:

The presence of FXF sites in N-Myc resembles sites for ERK2 and JNK interactions as shown in Orand & Jensen 2025; Kragelj Biomolecules 2021; Hendus-Altenburger 2018 and more. How does these sites in N-Myc compare to the FXFP sites for MAPK interactors generally observed, should be discussed (described by the Kornfeld and the Ghoose groups in the 90'ies). A discussion on motifs is missing.

Response:

As AurA is not a MAP kinase, N-Myc is not a substrate of AurA, and MAPKs to our knowledge are not active on N-Myc, a detailed comparison with described features of MAPK interactors, we find it difficult to see how such a comparison would increase the understanding of the N-Myc-AurA system in the context of this study. However, we have shown earlier that around 50% of more than 350 interactors depend on one or several conserved so-called MYC boxes (Kalkat et al., Mol Cell 2018; Figure 1). Beyond these conserved MYC boxes, comprising 20-50 residues each, interaction patterns/motifs in MYC are not yet resolved, and it is still very open whether the MYC boxes comprise embedded SLIM-type motifs or if interactions rely on entire or multiple MYC boxes. A comparative discussion on such specific motifs is therefore difficult to make.

However, we still agree with Reviewer #3 that it is relevant to discuss and compare binding motifs. To this end we compared the N-Myc binding regions identified in this study with Aurora binding regions identified in TPX2 (AurA), INCENP (AurB) and CEP192 (AurA) co-crystal structures. Like in our study, their interacting regions all feature aromatic/aliphatic residues flanked mainly by charged residues, most often negatively charged which makes sense as the N-lobe of the Auroras is highly positively charged. However, we could not identify any unifying, distinct SLIM-like motifs common to these interactors and N-Myc. Obviously this needs more structural and dynamic data to be completely evaluated. We still briefly mention this comparison in the discussion in the context of contextual modulation and competitive binding,

together with the need for further mapping of the structural and dynamic diversity of these complexes.

Reviewer #3 Comment 11:

Targeting kinases have shown quite dramatic side effects as these molecules have multiple partners. The suggestion of targeting the aromatic pockets to inhibit binding of N-Myc will also affect all other partners binding to this site. Thus, there are expectedly several side effects associated with such a drug scheme. Should be considered in the discussion (see e.g. work on JAK inhibitors)

Response:

We appreciate that Reviewer#3 is already considering implementations of drug design based on our current model study. Current drug targeting efforts of AurA has so far been limited to direct, ATP-like active-site inhibitors disrupting the enzymatic function of AurA. As described and referenced in the manuscript, although such inhibitors have shown certain effects in preclinical studies, their efficacy in clinical trials is poor, and it appears that simply blocking Aurora kinase activity is usually not enough to achieve maximum therapeutic efficacy. Alternative strategies to target AurA therefore need to be explored, including targeting of other regulatory interaction sites.

Although our study bears promise towards this direction, we believe that it is way too early to discuss pros/cons of potential future drug schemes in more detail here. Evaluating such an inhibitor for cross-reactivity to other kinases would be a classic step in kinase inhibitor drug development. Only when such an inhibitor is developed, can this specificity issue be addressed.

Reviewer #3 Comment 12:

The authors used the term “touchpoint” to describe the interaction. It is not clear what defines a touchpoint - is this an accepted term? They go on to mention “dynamic touchpoints”, how are they different from touchpoints? Why not use commonly used descriptions of interactions?

Response:

We appreciate the reviewer’s concern for clarity and have revised to commonly used descriptions of interaction depending on context.

Reviewer #3 Comment 13:

The abstract states this study goes against the “current dogma” – but the current dogma is not clear? Several disordered complexes involving IDPs have been described, and the current view is that retained disordered in complexes covers a continuum from fully folded upon binding to fully disordered. It requires some explanation and definitions to follow what the authors mean regarding “current dogma”.

Response:

Again, we apologize for lack of clarity. We used “current dogma” in the previous version related to the previous model of N-Myc-AurA interaction, which confines this to the N-Myc_{C61-89} segment. This model has become somewhat of a “dogma” as it led to a series of publications focused on interaction to this region, and our results go against this model in several ways. We have now revised our wording regarding this to exclude the wording “current dogma” and instead highlight it in the discussion as an alternative model that may be relevant in specific situations as part of a contextual modulation by multiple binding sites on N-Myc and AurA.

Reviewer #3 Minors:

It is concluded that the stability of the complex correlate with affinity, but no correlation plots or statistics have been done to support this. Can this be included to support this conclusion? Is unfolding reversible?

- In the first version, we did not include a correlation plot as the data points (# mutations, constructs) are too few to support any statistical evaluation. However, with the data we have, the correlation is close to linear and is now included in Main for completeness. In the nanoDSF setup, unfolding of AurA is not reversible, which is commonly observed in this setup, in contrast to a DSC experiment optimised to obtain equilibrium at each step of the denaturation and with optimized cell surfaces to resist aggregation.

The sequence in Fig. S3 should be included in the main figures, avoid spaces in the sequence, this is confusing.

- In the reviewed manuscript, the sequence in Supplementary Fig. 3 is that of N-Myc whereas the HDX-MS experiment in Main Figure - which we assume the Reviewer is referring to - instead shows the AurA protection pattern, where, due to space constraints, only amino acid numbers and secondary structure elements of AurA is included. With the new HDX-MS experiment performed to target N-Myc protection (see above), both Fig. 5 and Supplementary Fig. 6 in the revised manuscript have been remade with special attention to clarity.

Figure 5C, move data to SI and show the bar graphs with uncertainties instead.

Number of repetitions of the ITC should be at least $n=3$. Consider the numbers and the relevance (e.g. 3.98 ± 0.45 should be 4 ± 0.5). Uncertainties in the ITC should be on the figures

- We have now contributed triplicates of the ITC measurements in Fig. 5C, moved the data to Supplementary Fig. 9 and report the requested bar graph. Regarding accuracy, we have revised to two significant digits for all reported ITC parameters.

Why are both DDT and TCEP added?

- Both TCEP and DDT are strong reducing agents. We use both to take advantage of the stability of TCEP, making it suitable for long-term measurements and storage, while the less bulky structure of DDT can sometimes allow for more efficiency. DDT is also more cost efficient.

More description of the kinase assay is needed to understand the experiment

- The kinase assay, which is a commercial assay extensively used in the field, is now described in detail in Methods; see also our response above.

Post transcriptional versus post translational – why use these two terms?

- We agree and have revised to post-translational throughout.

Line 173, some underline under Che should be removed

Line 183, what is residue motifs?

Line 477 a-> alpha

Line 478 (REF Diana)

- These unclarities have now been revised.

REVIEWER COMMENTS

Point-2-point response on review after first revision

Reviewer #1 (Remarks to the Author):

The authors have thoroughly addressed all of my comments in this revision. The manuscript overall significantly advances our understanding of the interaction mode between AurA and N-Myc, and is suitable for publication in its current form.

Authors response:

We thank Reviewer #1 for the kind appreciation of our work, and for insightful and helpful suggestions during the review process.

Reviewer #3 (Remarks to the Author):

Hultman and coworkers have presented a revised manuscript where they have gone far to provide the additional experimental data requested which aide to support their conclusions. I appreciate the responsiveness of the authors to the suggestions made and in clarifying the text, not overinterpreting their findings. The manuscript is clearly improved with the new data, and with the attention to the textual formulation. However, I find that there are still a few issues and comments remaining that must be addressed:

Reviewer #3 Comment 1:

There is some inconsistency in the SEC-SAXS and the ITC, that is not reflected upon. While SEC SAXS suggest a 1:1 complex, the ITC is probing a 2:1 complex (molar ratio 0.5). Weak binding as 5 micromolar are often difficult to capture in co-elution on the SEC, so what is the origin of this discrepancy? Other disordered complexes form higher order complexes, and this may be very relevant to address here as well. Some explanation is warranted.

Authors response:

We acknowledge that applying the 1:1 ITC model is a first approximative description of the landscape of bound states, and that the ITC N-value ranges from 0.45 to 0.7 even with our extensive experimental optimisation. The N-value in ITC only properly reflects stoichiometry if the number of distinct binding events is identical to those reflected in the model. If we have multiple binding sites of varying affinities and thermodynamic profiles (low/high ΔH) in the titrating molecule, the N value will be lower than 1 when fitting a 1:1 model to the data, as additional sites will then contribute to early saturation (see for example Watson et al., JMB 2022). This agrees with the presence of several binding regions as suggested from our NMR and HDX data. More complex multisite models in the ITC fitting slightly increase the total N-value (N_1+N_2 etc), but many of the parameters are then poorly determined. As not to overinterpret our data, and to allow direct comparisons between constructs that may contain different numbers of binding regions, we have applied the 1:1 model to all our ITC data.

Our combined data suggests an interaction involving multiple binding sites on both interaction partners, however regarding stoichiometry, this can still be 1:1, as any or all sites can be occupied by the same receptor at any given time, even if the affinity of each site is weak. This would be consistent with co-elution on a SEC column, which could be facilitated by the multiple interaction sites maintaining complex formation despite the relatively low affinities. As reported by us in detail in Revision#1, all our stoichiometric data (SAXS, SEC-MALS) are consistent with a 1:1 stoichiometry, with no evidence of higher stoichiometries such as 2:1. Therefore, our data gives us no scientific support to suggest higher AurA occupancy than 1:1.

Action:

In Results, we note the lower-than-one N-value and provide a short explanation based on multiple binding regions, with reference to Watson et al JMB 2022. We also explain how this is consistent with co-elution on SEC. In Methods, we include and describe our choice of binding model with motivation as above.

Reviewer #3 comment 2:

Can NMR data help here, in the sense that the peak movements – even at the border of the disappearing regions – may respond differently at the different stoichiometric ratios. This will add to the complexity of the binding mode, but given the disordered nature of it, perhaps not too surprising.

Authors response:

While assessing the N-Myc- AurA interaction by NMR we lose peak intensity at very low stoichiometries in a non-recoverable manner for residues both in and adjacent to the binding region, with intensity remaining only at the N- and C-termini at 1:1 (see Revision #1). As this limits the range of stoichiometries that we can access by NMR, we cannot confidently refine our binding model to higher degrees of complexity using peak movements at this stage.

Reviewer #3 Comment 3:

If you have multiple binding sites in N-Myc that are non-identical, binding to multiple non-identical sites on Aurora A (discussion p. 17, line 508), is it then allovalency? (see e.g. Locasale, Allovalency revisited: an analysis of multisite phosphorylation and substrate rebinding. J Chem Phys 128:115106. doi:10.1063/1.2841124). In my opinion, and according to the definitions, it is not. It appears instead to be a case of multivalency. Please reconsider and relate to the definitions of different binding mechanism described in the literature.

Authors response:

In Revision #1, as requested by Reviewers #2 and #3, we paid special attention to revise wording referring to dynamic properties of the complex. We agree that definition of allovalency - with multiple identical binding sites - is not the case here although all binding sites include aromatic residues that affect binding when mutated. We have therefore now

also removed the word “allovalent” from the mentioned sentence in the discussion, which is the only time this word is used in the manuscript.

How to describe and understand IDP interactions is vividly discussed in the field and the literature classifying different binding models is vast and diverse. Within the scope and space of this article, we do not have space to include specific descriptions of the definitions, but refer to recent, comprehensive work to clarify these concepts for IDPs (Holehouse and Kragelund, Nature Rev Mol Cell Biol 2024). “Multivalency” within the IDP field is currently mainly used in the context of phase separation. The “fuzzy” definition broadly encompasses much of the IDP binding events but has little mechanistic detail. Within the limits of our experimental data, the N-Myc-AurA complex can be described as multivalent with high dynamics, but it could just as well be described as fuzzy. As we do not yet have any direct kinetic data to inform us about the specific binding mechanism, we prefer not to introduce a mechanistic discussion in this work as this would risk overinterpretation of our current data.

Action:

In the discussion we summarise our experimental data and conclude that “all these results hint at a mode of interaction where several regions in N-Myc contact multiple binding sites on AurA in a dynamic manner, resulting in a complex with multivalent and/or fuzzy properties.”

Reviewer #3 Comment 3:

It seems that the region of MB0 needs adjusting based on the R2s and on the I/I0 – it appears to extend on the C-terminal side, please rethink this region.

Response and action:

As for MYC regions MBI-MBIV, the MB0 region is defined based on the conservation between the different myc family proteins and their participation in protein interactions (Lourenco et al., Nat Rev Cancer, 2021; Kalkat et al., Mol Cell 2018). MB0 is thus not defined by the Aurora A interaction site. To support the reader, we have included a sequence alignment in Suppl Fig 1 where MB0 and MBI regions are annotated in detail.

Reviewer #3 Comment 4:

Are the R2 changes “dramatic” – or just what is expected from binding to a larger kinase? Please rethink.

Authors response:

The word “dramatic” was not used in the manuscript, only in the Reviewer response in Revision#1 to describe the fast increase in R2 relative to the amount of added AurA. We agree that this wording was unscientific, and we apologize.

In response to the Reviewer’s concerns, it is indeed reasonable to expect that the R₂ values in the bound state are substantially higher than in free N-Myc, due to the anticipated

increase in correlation time associated with the molecular weight (M_w) of the kinase. Comparing with for example Delaforge et al (JACS 2018), they determined $R_{2, \text{bound}}$ rates of 60–200 s^{-1} in a system size wise similar to ours.

If we assume that the observed increase in relaxation rate is solely attributable to the higher M_w of the IDP–kinase complex and not to conformational exchange between free and bound states or to internal dynamics within the bound state, then the loss of signal intensity should scale proportionally with the amount of added kinase. However, this is not what we observe: for some residues, 100% of the signal is lost at only 10% kinase added, which is a more unproportional response than what was reported by Delaforge et al. This suggests a significant, additional contribution to R_2 by fast exchange and/or multiple binding modes, although our current data cannot distinguish between these possibilities. A direct comparison between our data and the data in Delaforge et al is further challenged by different experimental conditions, such as different magnetic field strengths and temperatures; moreover, exchange rates are likely to differ between these two systems. Still, we can convincingly conclude from our NMR analysis that three distinct regions in N-Myc are similarly perturbed by AurA binding, as shown by CSPs, line broadening, and increased R_2 values.

Action:

We have clarified language in the Results section “Three distinct N-Myc regions interact with AurA” to ensure that the Reader notices that several factors contribute to line broadening and increased R_2 : correlation time, conformational exchange between free/bound and exchange processes in the bound state.